# Flow Matching in the Low-Noise Regime: Pathologies and a Contrastive Remedy

## Abstract

Flow matching has recently emerged as a powerful alternative to diffusion models, providing a continuous-time formulation for generative modeling and representation learning. Recent progress in generative visual foundation models suggests that flow-matching models could serve not only as generators but also as unified backbones for downstream discriminative tasks. This raises a natural question: can flow matching reliably learn high-quality representations directly from clean data? Yet, we show that this framework suffers from a fundamental instability in the low-noise regime. As noise levels approach zero, arbitrarily small perturbations in the input can induce large variations in the velocity target, causing the condition number of the learning problem to diverge. This ill-conditioning not only slows optimization but also forces the encoder to reallocate its limited Jacobian capacity toward noise directions, thereby degrading semantic representations. We provide the first theoretical analysis of this phenomenon, which we term the **low-noise pathology**, establishing its intrinsic link to the structure of the flow-matching objective. Our analysis reveals that this pathology creates a major bottleneck for using flow matching as a viable representation-learning framework, limiting its suitability as a discriminative visual backbone. Building on these insights, we propose **Local Contrastive Flow** (LCF), a hybrid training protocol that replaces direct velocity regression with contrastive feature alignment at small noise levels, while retaining standard flow matching at moderate and high noise. Empirically, LCF not only improves convergence speed but also stabilizes representation quality. These results suggest that addressing the low-noise pathology is essential for advancing flow-matching models toward unified generative–discriminative visual foundation models.

## 1 Introduction

Generative models (Ho et al. (2020); Song et al. (2020); Lipman et al. (2022); Liu et al. (2022)) have become central to modern machine learning, delivering state-of-the-art performance in tasks ranging from image (Esser et al. (2024)) and speech synthesis (Liu et al. (2023)) to molecular design (Irwin et al. (2024)) and scientific simulation (Wildberger et al. (2023)). Models such as diffusion models and flow models not only excel at producing high-fidelity samples but also offer the intriguing promise of learning representations that capture the underlying semantic structure of data (Li et al. (2023); Xiang et al. (2023b); Chen et al. (2024)). This dual capability has fueled growing interest in leveraging generative models as universal tools for both data generation and representation learning, bridging synthesis and understanding within a unified framework. Recent progress in visual foundation models has further intensified this interest( Chen et al. (2024); Liu et al. (2025); Fuest et al. (2024)), since models capable of both generation and discrimination provide a pathway toward unified backbones that support editing, perception, and high-level reasoning within a single architecture.

Flow models, in particular, provide a principled and efficient alternative to score-based diffusion models, enabling direct learning of continuous dynamics that map simple distributions to complex data. A key feature of these models is their training objective, which enforces consistency of the predicted velocity fields across multiple noise scales. This multi-scale supervision not only stabilizes training but also enforces semantic consistency of the data across different variations. Such a formulation naturally aligns with the objectives of representation learning (Chen et al. (2020);

Hadsell et al. (2006); He et al. (2020)), prompting a fundamental question: *Can generative models reliably learn useful representations directly from clean data*? Intuitively, one might expect that the cleaner the data, the easier it should be for a model to extract semantic information. Clean data lies close to the true data manifold, free of artificial distortions, and is often the most desirable input for downstream tasks such as classification, clustering, and segmentation. This question has become increasingly important as recent studies have suggested that diffusion and flow-based generative models may serve as promising representation learners for future visual foundation models.

Surprisingly, however, our empirical evidence suggests a starkly different reality. We evaluated the target loss and representation quality under different noise intervals. As shown in Figure 1, when training on samples with very low noise levels, which correspond to small time values in flow-based models, generative models often encounter significant learning difficulties. The loss curves exhibit poor convergence behavior, and the quality of the learned representations is substantially lower than that obtained at moderate noise levels. This occurs precisely in the regime where the input data are nearly identical to the original clean examples, creating an apparent paradox: *why does training become more difficult as data approaches its clean form?* This paradox directly challenges the use of flow matching as a practical representation-learning framework and raises doubts about its suitability as a unified visual backbone.

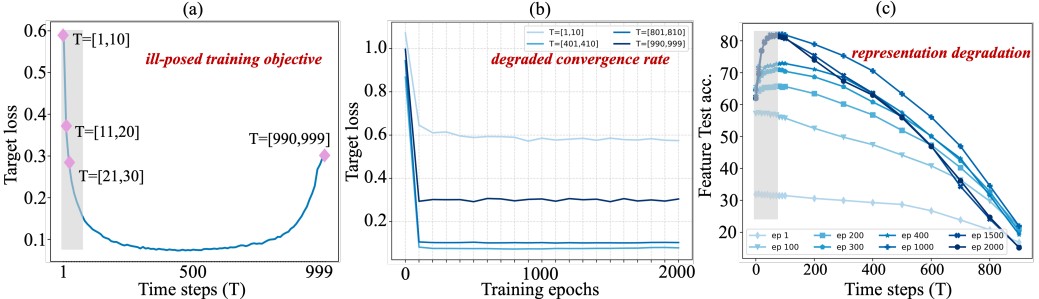

Figure 1: **Low-noise pathology in flow matching. (a) Ill-posed objective:** the objective loss in low-noise regions significantly larger than at moderate or high noise levels, even compared to fully noisy inputs. **(b) Degraded convergence:** gradient-based optimization converges extremely slowly in the low-noise regime, so that training objective at small $t$ dominate the overall loss landscape and become the primary bottleneck in the later stages of training. **(c) Representation degradation:** representation quality does not decrease monotonically with $t$, exhibiting an anomalous degradation under extremely low-noise regime. We train on CIFAR-10 and evaluate using the test set.

This phenomenon, which we term the **low-noise pathology**, is the central focus of our work. In rectified flow models, data samples are interpolated between clean data and random noise through a time-dependent process defined as $x_t = (1 - t) \cdot x_0 + t \cdot \epsilon$. As $t$ decreases toward zero, the supervision targets, such as the vector field $x_0 - \epsilon$, remain large and highly variable, while the differences between inputs simultaneously shrink because the contribution of $\epsilon$ to $x_t$ vanishes. This imbalance leads to a profound mismatch between the scale of input perturbations and the magnitude of the outputs that the model is required to predict. Crucially, this mismatch is not an artifact of architecture or implementation, but a structural property of the flow-matching objective itself.

We demonstrate that this mismatch induces a sharp increase in the condition number (Belsley et al. (1980); Pesaran (2015)) of the learning problem as the noise level decreases. In particular, we prove that the ratio between the magnitude of the output variations required by the model and the corresponding input differences scales inversely with the noise parameter $t$. As $t \to 0$, this ratio diverges, resulting in an increasingly ill-conditioned loss landscape that severely impedes optimization. This phenomenon offers a theoretical explanation for the empirical difficulties observed in the low-noise regime. Furthermore, the instability extends beyond optimization: it undermines the quality of learned representations. Features extracted under these conditions often degenerate, losing their semantic discriminability and thus their utility for downstream tasks. This theoretical insight shows that improving flow matching in the low-noise regime is essential not only for optimization, but also for enabling flow models to function as reliable discriminative visual backbones.

To address these issues, we present the first rigorous theoretical analysis of the low-noise pathology in flow models. We trace the root of this instability to the statistical properties of the target

vector field, revealing that such numerical issues emerge as an inherent consequence of the generative modeling objective itself. Building on these insights, we propose a novel training method termed **Local Contrastive Flow**. In contrast to conventional flow matching approaches that apply direct supervision at extremely low noise levels, our method adopts a contrastive learning paradigm. Representations obtained at moderate noise levels, where training dynamics are more stable and semantic information is better preserved, are used as anchors. The model is trained to align the representations of slightly perturbed versions of the same input with these anchors, while simultaneously encouraging greater separation between representations of different inputs. Since training at moderate noise levels preserves semantic structure without inducing numerical instability, this alignment enables the transfer of robust semantic features to representations at lower noise levels. As a result, LCF not only accelerates the training process but also improves the semantic quality of the learned representations, enhancing their effectiveness in downstream tasks.

We validate our approach through comparative experiments, demonstrating that LCF significantly improves both convergence speed and the quality of learned representations in flow models. Our results highlight that understanding and resolving the low-noise pathology is critical for unlocking the full potential of generative models as tools for both data generation and representation learning.

In summary, our contributions are as follows:

• We identify and formalize the low-noise pathology in flow models, revealing that such numerical issues emerge as an inherent consequence of the generative modeling objective itself.

• We further demonstrate how the low-noise pathology undermines the quality of learned representations, expanding the understanding of low-noise pathology beyond numerical optimization to representational effectiveness.

• We propose Local Contrastive Flow, a principled approach to accelerate generative training while alleviating the low-noise pathology.

## 2 BACKGROUND

Recent advances in large multimodal foundation models have pushed the frontier of unifying generation and understanding within a single architecture. Inspired by large language models such as GPT-4o (OpenAI (2024)) and Gemini 2.0 (DeepMind (2024)), recent efforts including Emu (Sun et al. (2023)), Chameleon (Team (2024)), and Metamorph (Tong et al. (2024)) introduce early-fusion strategies and unified token spaces to support cross-modal pretraining. Instruction-tuned variants such as Emu3 (Wang et al. (2024)) and Janus (Wu et al. (2025a)), together with scaling approaches in Janus-pro (Chen et al. (2025)) and Show-o (Xie et al. (2024)), highlight that task alignment and large-scale training are central to improving both reasoning and generative capabilities. Despite these empirical successes, our theoretical understanding of how such models jointly acquire generative and discriminative skills remains limited.

In parallel, diffusion-based generative models have demonstrated strong representational power beyond data synthesis. A growing body of work (Xiang et al. (2023a; 2025); Wu et al. (2025b); Yu et al. (2024); Wang & He (2025a); Li et al. (2025); Agarwal et al. (2025); Zhang et al. (2023)) shows that denoising objectives can yield transferable, semantically structured embeddings. Crucially, training under moderate noise levels has been observed to preserve feature continuity, whereas very low noise often leads to degraded representations (Pavlova & Wei (2025)). These findings suggest that generative supervision acts not only as a synthesis signal but also as an implicit representation-learning objective, raising questions about how this objective shapes feature geometry and generalization.

Taken together, these two perspectives motivate our study of flow matching in the low-noise regime. While unified models rely on generative supervision to balance synthesis and understanding, the low-noise pathology threatens this balance by destabilizing both optimization and representation quality. Our work provides a theoretical framework for analyzing this issue and introduces a practical remedy that preserves generative fidelity while stabilizing learned representations.

## 3 PRELIMINARIES

Flow matching (Lipman et al. (2022); Liu et al. (2022)) provides a continuous-time framework for generative modeling in which data samples are gradually corrupted by noise and then recovered by learning an instantaneous velocity field. Let $x_0 \sim p(x)$ be a data point and $\varepsilon \sim \mathcal{N}(0, I)$ independent noise. For $t \in [0, 1]$, define

$$x_t = \alpha_t\, x_0 + \beta_t\, \varepsilon, \quad \alpha_0 = 1,\ \beta_0 = 0, \quad \alpha_1 = 0,\ \beta_1 = 1, \tag{1}$$

where $\alpha_t$ decreases and $\beta_t$ increases in $t$. Under mild regularity, there exists a probability-flow ordinary differential equation (PF-ODE)

$$\dot{x}_t = v(x_t, t), \tag{2}$$

whose marginal at time $t$ matches the law of the interpolation $x_t$. In practice, one approximates the velocity field

$$v(x, t) = \mathbb{E}\left[\dot{x}_t \mid x_t = x\right] = \alpha_t'\, \mathbb{E}\left[x_0 \mid x_t = x\right] + \beta_t'\, \mathbb{E}\left[\varepsilon \mid x_t = x\right], \tag{3}$$

by a neural network $v_\theta(x, t)$. Training proceeds by minimizing the mean-squared velocity matching loss

$$\mathcal{L}_{\text{flow}}(\theta) = \mathbb{E}_{x_0, \varepsilon, t} \left\| v_\theta(x_t, t) - v^*(x_t, t) \right\|^2, \tag{4}$$

where $v^*(x_t, t) = \alpha_t'\, x_0 + \beta_t'\, \varepsilon$ and $t$ is drawn (e.g., uniformly) from $[0, 1]$. Under sufficient model capacity, minimizing $\mathcal{L}_{\text{flow}}$ yields a vector field whose backward integration from pure Gaussian noise at $t = 1$ to $t = 0$ recovers high-fidelity samples from the data distribution $p(x)$.

When desired, one may equivalently generate samples via the reverse stochastic differential equation (SDE)

$$\mathrm{d}x_t = v_\theta(x_t, t)\, \mathrm{d}t - \frac{1}{2}\omega_t\, s(x_t, t)\, \mathrm{d}t + \sqrt{\omega_t}\, \mathrm{d}\bar{W}_t, \tag{5}$$

where $s(x, t) = -\sigma_t^{-1} \mathbb{E}\left[\varepsilon \mid x_t = x\right]$ is the score function and $\omega_t$ a suitable diffusion coefficient (Ma et al. (2024)). In either ODE or SDE form, the flow-matching framework elegantly unifies score-based and diffusion-based generative modeling, while enabling efficient likelihood evaluation and sampling via off-the-shelf ODE/SDE solvers.

## 4 ILL-POSEDNESS IN THE LOW-NOISE REGIME

While flow matching offers a powerful framework for generative modeling, its behavior under low-noise conditions reveals a fundamental ill-posedness that hinders both optimization and representation learning. This section investigates how this pathology arises in the low noise regime, why it reflects an inherent limitation of the flow matching objective, and what implications it bears for optimization and generalization.

### 4.1 DIVERGENCE OF CONDITION NUMBER UNDER FLOW MATCHING

In statistical learning, the condition number measures how sensitive a function is to input perturbations, which is widely used to analyze optimization dynamics and generalization (Demmel (1987); Scaman et al. (2017); Ji et al. (2021)). We thus introduce the local condition number $\kappa = \frac{\|\Delta v\|}{\|\Delta x\|}$, along with Proposition 1.

**Proposition 1** (Divergent Condition Number). *Let $x_0 \in \mathbb{R}^d$ be fixed with $\|x_0\|_2 \le B$ for some constant $B > 0$. Consider the interpolation family $x_t = \alpha_t x_0 + \beta_t \varepsilon$, where $\alpha, \beta \in C^1([0, 1])$, $\alpha_0 = 1, \beta_0 = 0$, and for $t > 0$ we have $\beta_t > 0$. Fix two times $t_1, t_2 > 0$ and draw independent noise realizations $\varepsilon_1, \varepsilon_2 \sim \mathcal{N}(0, I_d)$. Let $\Delta v := v^\star(x_{t_1}^{(1)}, t_1) - v^\star(x_{t_2}^{(2)}, t_2), \Delta x := x_{t_1}^{(1)} - x_{t_2}^{(2)}$, where $x_{t_i}^{(i)} = \alpha_{t_i} x_0 + \beta_{t_i} \varepsilon_i$. If $t_1, t_2 \approx t$, then the condition number satisfies*

$$\kappa_E(t_1, t_2; x_0) := \frac{\sqrt{\mathbb{E}_{\varepsilon_1, \varepsilon_2} \|\Delta v\|_2^2}}{\sqrt{\mathbb{E}_{\varepsilon_1, \varepsilon_2} \|\Delta x\|_2^2}} \gtrsim \frac{\beta_t'}{\beta_t} \xrightarrow[t \downarrow 0]{} \infty. \tag{6}$$

This infinite ill-conditioning in Proposition 1 implies that arbitrarily small deviations in the noisy input can cause large output variations, rendering the inverse problem of estimating $v^*$ from $x_t$ severely ill-posed in the low-noise regime. Such ill-posedness has direct consequences for learning. Gradient-based optimization of the flow-matching objective relies on smooth variations of the velocity field over the data manifold. However, in the low-noise regime, the velocity target remains large and highly variable, while the differences between inputs simultaneously shrink. As a result, the gradient landscape becomes increasingly sharp and unstable, leading to degraded convergence rate.

## 4.2 Degraded convergence rate under low-noise regime

A central question in understanding the pathological behavior of flow matching in the low-noise regime is how the training dynamics slow down and eventually stagnate. While standard analyses guarantee polynomial or exponential convergence rates in terms of sample size and optimization steps, our results indicate that in the low-noise setting the convergence rate can degrade substantially due to intrinsic operator ill-conditioning.

The key observation is that infinite ill-conditioning implies that even infinitesimal perturbations in the noisy input $x_t$ can induce disproportionately large variations in the velocity target $v^*(x_t, t)$. This makes the regression problem of estimating $v^*$ from $x_t$ fundamentally unstable. Beyond this conceptual instability, optimization itself becomes adversely affected: the training landscape grows increasingly ill-conditioned as $t \to 0$, directly slowing down convergence. Subject to some necessary assumptions, we formalize this below.

**Proposition 2** (Linear Convergence of Gradient Descent (Karimi et al. (2016))). *Let $\mathcal{L}(\theta)$ be twice continuously differentiable in a neighborhood of its global minimizer $\theta^*$ and $\{\theta_k\}_{k \geq 0}$ evolve under $\theta_{k+1} = \theta_k - \eta \nabla \mathcal{L}(\theta_k)$ with $\eta \in (0, 2/L)$. Assume there exist constants $0 < \mu \leq L$ such that for all $\theta$ in that neighborhood $\mu I \preceq \nabla^2 \mathcal{L}(\theta) \preceq LI$. Then to achieve $\|\theta_k - \theta^*\|_2 \leq \varepsilon$ requires*

$$k \geq \Omega\left(\kappa \log \frac{1}{\varepsilon}\right), \tag{7}$$

*where $\kappa := L/\mu$ is the Hessian condition number.*

**Proposition 3** (Slow Convergence under Low-noise Regime). *Let $v_\theta(x, t)$ be a twice differentiable parametric model and consider the flow matching loss $\mathcal{L}_{flow}(\theta)$. Suppose in a neighborhood of a local minimizer $\theta^*$ the Gauss–Newton approximation is valid so that the Hessian satisfies $H(\theta^*) \approx 2\,\mathbb{E}[J_{\theta^*}^\top J_{\theta^*}]$, and assume model Jacobians are operator-norm bounded (finite sensitivity per parameter). Then there exist positive constants $C, c$ (depending on model Jacobian bounds and data moments) such that for sufficiently small $t$: $\kappa(H(\theta^*)) \geq C\left(\frac{\beta_t'}{\beta_t}\right)^2 - c$. Consequently, to reach $\|\theta_k - \theta^*\| \leq \varepsilon$, gradient descent requires at least*

$$k = \Omega\left(\left(\frac{\beta_t'}{\beta_t}\right)^2 \log \frac{1}{\varepsilon}\right). \tag{8}$$

Proposition 3 shows that as the noise parameters $t \to 0$, the effective condition number diverges, and the optimal convergence rate vanishes. This establishes a intuitive mechanism for *degraded convergence* in the low-noise regime: the optimization speed is fundamentally limited by operator ill-conditioning. In practice, this also explains why representations exhibit slower improvement, stagnation, and eventual degradation as training progresses under low noise.

## 4.3 Representation degradation under the low-noise regime

In this subsection we develop a principled account of *representation degradation* from a geometric degeneracy perspective to explaining why a degradation in representation quality occurs for small $t$.

**Setup and regularity.** We consider the exact instantaneous velocity is $v^*(x_t, t) = \alpha_t' x_0 + \beta_t' \varepsilon = (\beta_t'/\beta_t) x_t + (\alpha_t' - \alpha_t \beta_t'/\beta_t) x_0$, so the linear map from perturbations in $x_t$ to perturbations in $v^*$ is dominated by the scalar factor $\beta_t'/\beta_t = \Theta(1/t)$. Let $h = g_\ell(x)$ denote the representation at layer $\ell$, adn view the target mapping $x_t \mapsto v^*(x_t, t)$ as the linear operator (on perturbations)

$M_t \approx (\beta'_t/\beta_t)I_d$ plus lower–order terms. The model implements an approximant of $M_t$ via the Jacobian composition $\widehat{M_t} := J_u(h) J_g(x_t) = \frac{\partial u_\phi}{\partial h} \frac{\partial g_\ell}{\partial x}(x_t)$, whose operator norm is bounded by $L_u L_g$ from equation 9.

$$\|J_u(h)\|_{\mathrm{op}} \leq L_u, \qquad \|J_g(x)\|_{\mathrm{op}} \leq L_g. \tag{9}$$

(These caps model weight decay, layer normalization, or other effective capacity constraints.)

Intuitively, the encoder $g_\ell : \mathcal{X} \to \mathcal{H}$ endows the data manifold with a pullback metric $G(x_t) = J_g(x_t)^\top J_g(x_t)$ on the input space. Directions in input space that are *collapsed* by the encoder correspond to small singular values of $J_g(x_t)$ and small eigenvalues of $G(x_t)$. Discriminative tasks rely on preserving certain manifold directions (those that separate classes or semantics). The flow–matching target, however, demands strong sensitivity in the directions where $M_t$ has large action. When the demanded sensitivity of $M_t$ outstrips the maximal implementable sensitivity of the composed Jacobian $J_u J_g$, the encoder must concentrate its limited Jacobian norm on the directions that reduce the main loss contribution; this necessarily reduces sensitivity in other (semantic) directions, producing a form of *degeneracy* that degrades representation quality.

**Proposition 4** (Necessary Jacobian reallocation under high target gain). *Let $S \subset \mathbb{R}^d$ be a linear subspace (the "noise subspace") and denote by $P_S$ the orthogonal projector onto $S$. Let the approximation residual be $r_t := \| P_S(M_t - \widehat{M_t})\|_{\mathrm{op}}$ (the operator norm of the projected residual onto $S$). Then the encoder Jacobian satisfies the lower bound*

$$\sup_{\substack{v \in S \\ \|v\|_2=1}} \|J_g(x_t)\, v\|_2 \;\geq\; \frac{\|P_S M_t\|_{\mathrm{op}} - r_t}{L_u}. \tag{10}$$

*In words: matching the high sensitivity demanded by $M_t$ forces the encoder Jacobian to increase its gain along some noise direction.*

The above lower bound shows that, as $\|P_S M_t\|$ grows (for typical schedules this growth is $\Theta(\beta'_t/\beta_t)$), the encoder must increase its Jacobian gain along some directions. Since the encoder Jacobian operator norm is globally capped by $L_g$, this increased gain comes at the expense of *other* input directions. The encoder's Jacobian singular value spectrum must redistribute mass, reducing sensitivity along some semantic directions and creating a degeneracy in the pullback metric.

**Proposition 5** (Representation degradation under high target gain). *Assume input space decomposes orthogonally as $\mathbb{R}^d = S_{\mathrm{sem}} \oplus S_{\mathrm{noise}}$ with $\dim S_{\mathrm{sem}} = r$. Class differences $\Delta x_{c,c'} = x_0^{(c)} - x_0^{(c')} \in S_{\mathrm{sem}}$ and $\delta_0 = \min_{c \neq c'} \|\Delta x_{c,c'}\|_2 > 0$. Define the required noise–direction gain $g_{\mathrm{req}}(t) := \frac{\|P_{S_{\mathrm{noise}}} M_t\|_{\mathrm{op}} - r_t}{L_u}$, and assume $g_{\mathrm{req}}(t) > 0$, $Q(t) = \min_{c \neq c'} \|g_\ell(x_t^{(c)}) - g_\ell(x_t^{(c')})\|_2$ (encoded class separation). Then:*

$$Q(t) \;\leq\; \sqrt{B^2 - g_{\mathrm{req}}(t)^2}\; \delta_{\max}. \tag{11}$$

*Therefore, as $g_{\mathrm{req}}(t) \uparrow$ (for instance when $\|P_{S_{\mathrm{noise}}} M_t\|_{\mathrm{op}}$ grows like $\beta'_t/\beta_t$), the upper bound equation 11 decreases; when $g_{\mathrm{req}}(t)^2 \geq B^2$ the right-hand side is zero and encoded semantic separation can be forced to (near) zero.*

Propositions 4 and 5 together show that when the target Jacobian's demand in certain (noise) directions exceeds the model's implementable Jacobian gain, the encoder is forced to reallocate its finite Jacobian capacity toward those noise directions. Because capacity is redistributed (not created), sensitivity in semantic directions falls: the pullback metric degenerates along task-relevant axes and downstream discriminative performance degrades. This yields the observed nonmonotonic pattern of representation quality as $t$ decreases: initially, moderate increases in sensitivity help, but once $M_t$'s demand passes a threshold the encoder sacrifices semantic directions and representation quality falls.

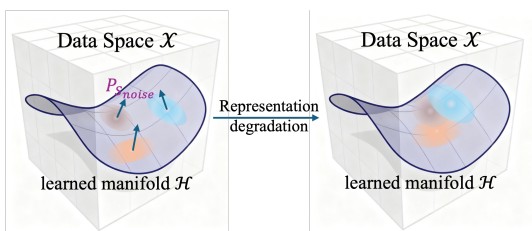

Figure 2: Representation degradation in low-noise regime: The learned manifold $\mathcal{H}$ loses semantic discriminability as noise level $t \to 0$.

The theoretical insights further suggest concrete strategies for stabilizing training. Methods that avoid or regularize the problematic low-noise regime, such as truncating the training objective to $t \geq T_{\min}$, can directly counteract covariance collapse. Similarly, contrastive regularizers that enforce separation between features at small $t$ and their counterparts at moderate $t$ act as information-preserving constraints, preventing rank deficiency in the feature covariance. In all cases, the guiding principle is to mitigate the condition-number explosion by either bypassing the low-noise region or explicitly preserving the variance structure of intermediate features.

## 5 METHOD: LOCAL CONTRASTIVE FLOW

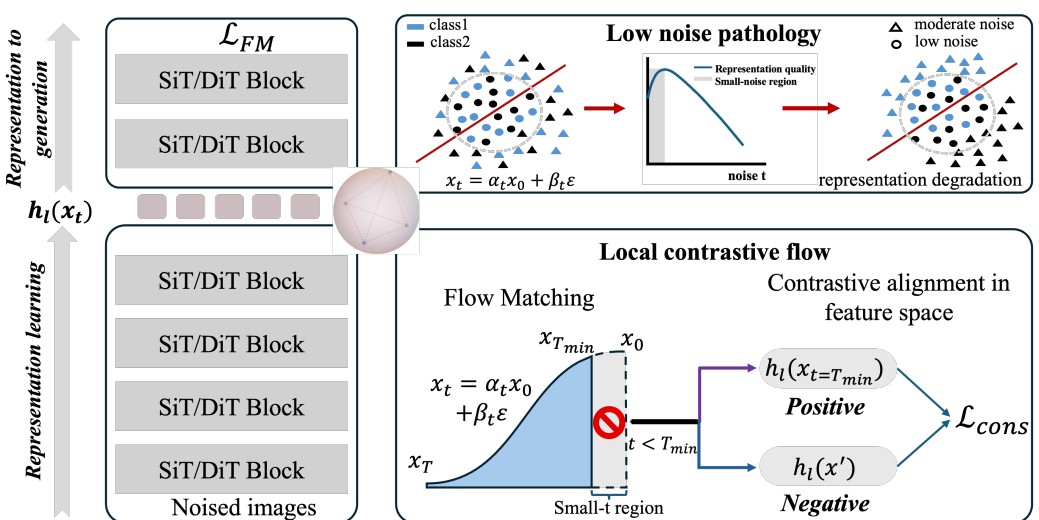

Figure 3: Overview of Local Contrastive Flow (LCF). At moderate and high noise levels, standard flow matching is applied to learn the velocity field. For small noise ($t < T_{\min}$), anchors at $t = T_{\min}$ provide positive targets, and contrastive alignment prevents representation degradation and ill-conditioning.

To overcome the ill-conditioning and representation degradation that afflict flow-matching models at very low noise levels, we introduce **Local Contrastive Flow**. As illustrated in Figure 3, Local Contrastive Flow partitions training into two regimes. For $t \geq T_{\min}$, we retain standard flow matching to preserve accurate velocity regression. For $t < T_{\min}$, anchors computed at $t = T_{\min}$ serve as positive targets, and contrastive alignment with other batch representations stabilizes training and prevents representation degradation. This hybrid protocol ensures well-conditioned optimization across all noise levels.

Concretely, let $x_t = \alpha_t x_0 + \beta_t \varepsilon$ denote the interpolation from clean data $x_0$ to noise $\varepsilon$, and $v_\theta(x_t, t)$ our neural approximation to the true velocity $v^\star(x_t, t) = \alpha'_t x_0 + \beta'_t \varepsilon$. We choose a threshold $T_{\min} > 0$ so that for $t \geq T_{\min}$, $\beta_t$ remains bounded away from zero and $\beta'_t / \beta_t$ is finite. In this region we retain the usual mean-squared flow-matching objective, sampling $t$ uniformly and minimizing

$$\mathcal{L}_{\mathrm{FM}} = \mathbb{E}_{x_0, \, \varepsilon, \, t \geq T_{\min}} \left\| v_\theta(x_t, t) - v^\star(x_t, t) \right\|_2^2. \tag{12}$$

This ensures that for moderate and high noise levels, the model learns an accurate velocity field and preserves the well-conditioned behavior of standard flow matching.

For noise levels below $T_{\min}$, direct regression of $v^\star$ becomes numerically unstable as $\beta_t \to 0$. Instead, we exploit a contrastive feature alignment at an intermediate layer $\ell$. Specifically, for a batch $\{x_t^{(i)}\}_{i=1}^{B}$ with random time indices $t$, we designate as *anchors* only those representations corresponding to $t < T_{\min}$, i.e.

$$z^{(i)} = h_\ell\big(x_t^{(i)}\big), \quad t < T_{\min}, \tag{13}$$

where $h_\ell$ denotes the representation at layer $\ell$. For each anchor $z^{(i)}$, we define its positive sample as the corresponding noiseless embedding $h_\ell(x_{T_{\min}}^{(i)})$, and define the negatives as all other representations within the same batch, namely $\{h_\ell(x_t^{(j)}) : j \neq i\}$ with $t$ arbitrary. We detach all the contrastive samples from the computation graph, so that gradient updates affect only $z^{(i)}$, then yields the contrastive loss:

$$\mathcal{L}_{\text{cons}} = -\frac{1}{|\mathcal{A}|} \sum_{i \in \mathcal{A}} \log \frac{\exp\left(-\frac{1}{\tau} \|z^{(i)} - h_\ell(x_{T_{\min}}^{(i)})\|_2^2\right)}{\sum_{j \neq i}^{B} \exp\left(-\frac{1}{\tau} \|z^{(i)} - h_\ell(x_t^{(j)})\|_2^2\right)}, \tag{14}$$

where $\tau$ is a temperature parameter, $B$ the batch size, and $\mathcal{A}$ indexes the anchors with $t < T_{\min}$. This formulation ensures that only low–noise representations are explicitly aligned with their mild counterparts, while simultaneously pushing them away from all other batch samples across arbitrary noise levels, thereby alleviates the divergent conditioning that plagues the original velocity regression.

In practice, we interleave these two objectives into a single loss:

$$\mathcal{L}_{LCM}(\theta) = \mathcal{L}_{\text{FM}} + \lambda \mathbb{E}_{t < T_{\min}} \mathcal{L}_{\text{cons}}. \tag{15}$$

We seamlessly integrate the computation of positives by concatenating $x_{T_{\min}}^{(i)}$ within each batch. Since $T_{\min} \ll T$, it does not significantly increase the memory usage. The temperature $\tau$ and weighting $\lambda$ are chosen so that the magnitudes of the flow-matching and contrastive losses are comparable near $t = T_{\min}$, ensuring a smooth transition between regimes.

This Local Contrastive Flow thus preserves the high-noise fidelity of standard flow matching while eliminating the small-noise singularity, yielding stable training, well-conditioned feature representations, and high-quality generative samples.

## 6 EXPERIMENTS

### 6.1 EXPERIMENT SETUP

Our experimental evaluation is conducted using the unconditional DiT architecture (Peebles & Xie (2023)), which has recently demonstrated strong performance in generative modeling. We train models on two benchmark datasets of increasing complexity, CIFAR-10 $32 \times 32$ and Tiny-ImageNet $64 \times 64$, to enable controlled evaluation of generative performance and representation quality in both low- and medium-scale regimes. To ensure comparability across settings, we adopt consistent training protocols, including data preprocessing, optimization hyperparameters, and noise schedules aligned with flow matching. Unless otherwise stated, all models are trained from scratch without auxiliary pretraining. We evaluate models along two axes: (i) **Generative performance**, using Fréchet Inception Distance (FID) computed over 7.5k generated samples; and (ii) **Representation quality**, using linear probing accuracy of intermediate features on held-out test data. The linear probe classifier is trained for 15 epochs with Adam at learning rate $1 \times 10^{-3}$ and batchsize 128.

### 6.2 LOCAL CONTRASTIVE FLOW IMPROVES FLOW MATCHING

We now empirically validate the effectiveness of *Local Contrastive Flow* (LCF) in addressing the pathologies of flow matching identified in Section 4. Our experiments focus on two central questions: (i) can LCF mitigate representation degradation in the low-noise regime, and (ii) does LCF accelerate convergence and improve sample quality in generative modeling? We also provide a comparative analysis against related modifications including Dispersive Loss (Wang & He (2025b)), DDAE++ (Xiang et al. (2025)) and ablation study. Unless otherwise specified, we adopt $T_{min} = 20$ for CIFAR-10, $T_{min} = 100$ for Tiny-ImageNet, $\lambda = 1$ and $\tau = 0.5$, following the same training protocols as in the baseline FM models (Sun et al. (2025)).

**LCF alleviates representation degradation.** In Section 4.3, we established that the standard flow matching objective inevitably suffers from representation degradation under small noise, leading to non-monotonic representation quality across time steps. We now evaluate whether LCF can alleviate

this effect. To this end, we measure the quality of learned intermediate representations across different time steps $t$ using linear classification accuracy, following the methodology described in (Xiang et al. (2023b)). Figure 4 reports results on CIFAR-10 and Tiny-ImageNet.

In both datasets, baseline FM exhibits the characteristic degradation curve: the representation quality does not decrease monotonically with $t$; instead, it exhibits an anomalous degradation under extremely low-noise regime, thus showing an overall trend of first increasing and then decreasing. In contrast, LCF maintains substantially higher representation quality at small $t$, producing a smoother and more stable curve with delayed degradation. This observation indirectly corroborates the theoretical insight that local contrastive objectives act as a regularizer, redistributing Jacobian mass and preventing semantic information from being overwhelmed by noise-driven amplification.

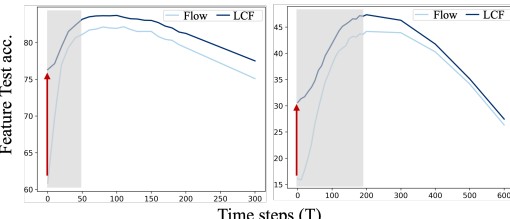

Figure 4: Feature test accuracy versus time steps (T) for Flow baseline and LCF. *Left*: Results on CIFAR-10. *Right*: Results on Tiny-ImageNet.

**LCF accelerates convergence and improves sample quality.** We next evaluate the impact of LCF on optimization dynamics. Proposition 3 predicts that the divergent condition number in FM slows convergence in the low-noise regime, which should manifest in delayed improvements in sample quality. To verify this, we track Fréchet Inception Distance (FID) during training. Figure 5 presents FID curves on CIFAR-10 and Tiny-ImageNet.

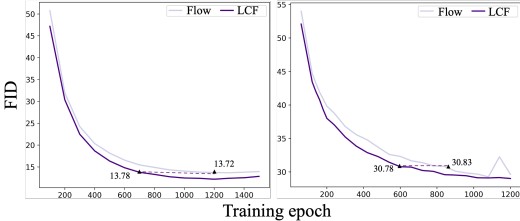

Figure 5: FID scores versus training epochs for Flow baseline and LCF. *Left*: Results on CIFAR-10. *Right*: Results on Tiny-ImageNet.

On both benchmarks, FM with LCF reaches a given FID threshold with significantly fewer iterations compared to standard FM, demonstrating accelerated convergence. Moreover, LCF consistently attains lower final FID, indicating that alleviating representation degradation not only accelerates training but also enhances generalization in the learned generative model.

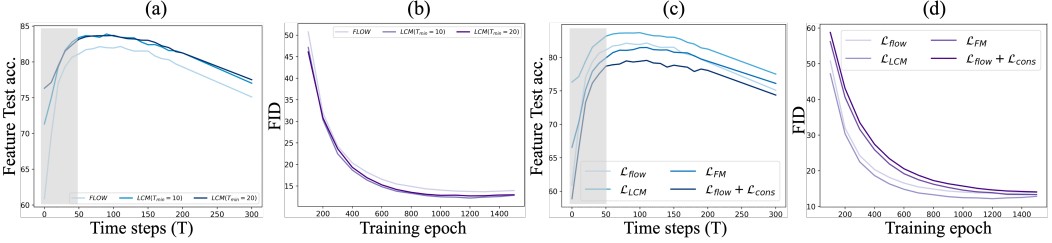

Figure 6: Ablation study of Local Contrastive Flow. (a)(b) Effect of different $T_{\min}$ choices. Larger $T_{\min}$, which uses higher-quality representations as positive anchors, reduces representation degradation but introduces fluctuations in generation quality, indicating a trade-off depending on the dataset. (c)(d) Effect of loss design. $\mathcal{L}_{\text{flow}}$ is the baseline; $\mathcal{L}_{\text{FM}}$ removes regression at small noise but lacks contrastive alignment; $\mathcal{L}_{\text{flow}} + \mathcal{L}_{\text{cons}}$ applies contrastive loss without stopping regression.

**Comparison with alternative remedies.** Finally, we compare LCF against several recent modifications of flow matching that also aim to bridge the gap between generative modeling and representation learning: *Dispersive Loss* and *DDAE++*. Figure 7 reports results on CIFAR-10 in terms of both representation quality and generative performance. From the results, both the dispersive loss and DDAE++ appear to yield only a slight improvement in generation quality. Dispersive loss shows no enhancement in representation quality. DDAE++ can improve the peak representation quality, but has no effect on representation degradation. In contrast, LCF achieves faster convergence and consistently stronger representations.

**Ablation Studies.** We further investigate the effect of key components in LCF. Figure 6(a)(b) shows results for different values of $T_{\min}$. Using higher-quality representations as positive anchors (larger $T_{\min}$) generally alleviates representation degradation more effectively, but we also observe fluctuations in generation quality, suggesting that the optimal choice of $T_{\min}$ may depend on the data distribution. In Figure 6(c)(d), we compare different loss formulations. Here, $\mathcal{L}_{\text{flow}}$ denotes the baseline, $\mathcal{L}_{\text{FM}}$ disables regression in the low-noise region but does not apply contrastive alignment,

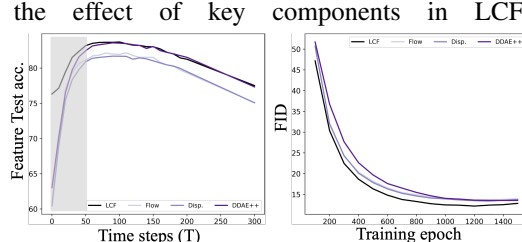

Figure 7: Comparison of generation quality and representation quality with dispersive loss and DDAE++.

and $\mathcal{L}_{\text{flow}} + \mathcal{L}_{\text{cons}}$ applies contrastive loss without stopping regression. The results confirm that simply removing regression at low noise is insufficient, while combining flow matching with contrastive alignment provides the most stable optimization and representation quality.

**Effect of timestep sampling schedule.** We also examine whether alternative timestep sampling strategies can alleviate the low-noise pathology. Following the Logit–Normal scheme of Esser et al. (2024),

$$\pi_{\ln}(t; m, s) = \frac{1}{s\sqrt{2\pi}} \frac{1}{t(1-t)} \exp\left(-\frac{(\text{logit}(t) - m)^2}{2s^2}\right), \tag{16}$$

we train three flow-matching models for 1200 epochs with parameters $m \in \{-0.5, 0, 0.5\}$ and $s = 1$. As shown in Figure 8 (left), all schedules exhibit a non-monotonic dependence of feature test accuracy on time step $T$, with a clear degradation toward the clean-data regime. The right panel of Figure 8 (right) visualizes the corresponding timestep densities: $m = -0.5$ concentrates training on low-noise timesteps, $m = 0$ favors higher noise, and $m = 0.5$ emphasizes intermediate noise.

We also reports the FID scores: 12.86 for $m = -0.5$, 17.50 for $m = 0.5$, and 19.47 for $m = 0$. These results reveal a tradeoff. Shifting probability mass away from the extreme low-noise region can somewhat improve the quality of clean-image representations, but typically degrades generative quality, whereas emphasizing low-noise timesteps improves FID yet leaves the clean-data representations suboptimal. In other words, adjusting the sampling distribution alone cannot simultaneously optimize both aspects. In contrast, Local Contrastive Flow explicitly reshapes the low-noise objective so that it improves representations at $t \approx 0$ while preserving strong generative performance, addressing this tradeoff more directly than schedule tuning.

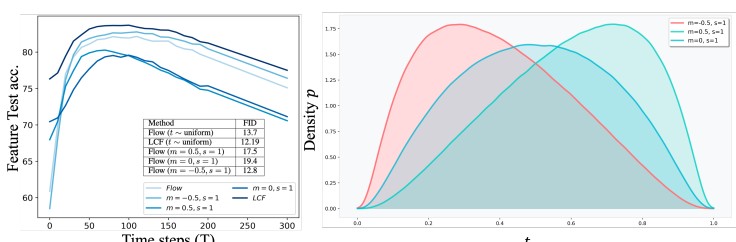

Figure 8: Effect of Logit–Normal timestep sampling on representation quality. Left: feature test accuracy across time steps for three schedules. Right: corresponding timestep densities.

# 7 CONCLUSION & FUTURE WORK

In this paper, we identified and analyzed a fundamental pathology of flow matching in the low-noise regime, where vanishing input perturbations induce disproportionately large changes in velocity targets, leading to divergent conditioning, slow convergence, and degraded representations. Our analysis and experiments on CIFAR-10 and Tiny-Imagenet serve as a proof of concept, demonstrating the theoretical validity and empirical feasibility of the approach in a controlled setting. Future directions include extending Local Contrastive Flow to high-resolution image and multimodal benchmarks, exploring adaptive schedules for anchor selection, and integrating contrastive alignment into large-scale training pipelines. We believe these steps will establish a foundation for applying flow matching reliably in real-world scenarios and for advancing unified generative–understanding architectures.

## REPRODUCIBILITY STATEMENT

To ensure the reproducibility of our work, we provide all necessary theoretical, experimental, and implementation details in the main text and Appendix. The theoretical analysis of the low-noise pathology, including assumptions, propositions, and proofs, is summarized in Section 4 and expanded in Appendix A.2. Details of the proposed Local Contrastive Flow method, including loss functions, training protocol, and implementation pseudo-code, are given in Section 5 and Appendix A.3. Experimental settings, hyperparameters, and evaluation metrics are specified in both the main paper and Appendix A.4, enabling faithful reproduction of reported results.

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

# A APPENDIX

## A.1 THE USE OF LARGE LANGUAGE MODELS

In preparing this manuscript, we employed a large language model (LLM) to assist with polishing the writing and improving grammatical clarity. The LLM was used exclusively for linguistic refinement, such as rephrasing sentences, smoothing transitions, and ensuring consistency in style, without altering the technical content, experimental results, or theoretical claims. All scientific contributions, mathematical derivations, and empirical findings remain the work of the authors.

## A.2 PROOFS AND ADDITIONAL THEORETICAL ANALYSIS

**Proposition** (1. Divergent Condition Number). *Let $x_0 \in \mathbb{R}^d$ be fixed with $\|x_0\|_2 \leq B$ for some constant $B > 0$. Consider the interpolation family $x_t = \alpha_t x_0 + \beta_t \varepsilon, \varepsilon \sim \mathcal{N}(0, I_d)$, where $\alpha, \beta \in C^1([0,1])$, $\alpha_0 = 1, \beta_0 = 0$, and for $t > 0$ we have $\beta_t > 0$. Define the instantaneous (ground truth) velocity $v^\star(x_t, t) = \alpha'_t x_0 + \beta'_t \varepsilon$. Fix two times $t_1, t_2 > 0$ and draw independent noise realizations $\varepsilon_1, \varepsilon_2 \sim \mathcal{N}(0, I_d)$. Let*

$$\Delta v := v^\star(x_{t_1}^{(1)}, t_1) - v^\star(x_{t_2}^{(2)}, t_2), \qquad \Delta x := x_{t_1}^{(1)} - x_{t_2}^{(2)}, \tag{17}$$

*where $x_{t_i}^{(i)} = \alpha_{t_i} x_0 + \beta_{t_i} \varepsilon_i$. Define the expected local condition ratio*

$$\kappa_E(t_1, t_2; x_0) := \frac{\sqrt{\mathbb{E}_{\varepsilon_1, \varepsilon_2} \|\Delta v\|_2^2}}{\sqrt{\mathbb{E}_{\varepsilon_1, \varepsilon_2} \|\Delta x\|_2^2}}. \tag{18}$$

*Assume additionally that there exist constants $c_1, c_2 > 0$ and an exponent $p > 0$ such that, for $t$ sufficiently small, $c_1 t^{p-1} \leq \frac{\beta'_t}{\beta_t} \leq c_2 t^{p-1}$.(In particular for a regular power law $\beta_t \asymp t^p$ one has $\beta'_t / \beta_t \asymp p/t$.)*

*Then the following lower bound holds for all sufficiently small $t_1, t_2 > 0$:*

$$\kappa_E(t_1, t_2; x_0) \geq \frac{\min(\beta'_{t_1}, \beta'_{t_2})}{\sqrt{\beta_{t_1}^2 + \beta_{t_2}^2 + \|x_0\|_2^2 (\alpha_{t_1} - \alpha_{t_2})^2 / d}}. \tag{19}$$

*Consequently, if $\beta_t \to 0$ and $\beta'_t / \beta_t \to \infty$ as $t \downarrow 0$ (e.g. $\beta_t \asymp t^p$ with $p > 0$), then for any sequence $t_n \downarrow 0$ we have*

$$\kappa_E(t_n, t_n; x_0) \gtrsim \frac{\beta'_{t_n}}{\beta_{t_n}} \xrightarrow[n \to \infty]{} \infty, \tag{20}$$

*i.e. the expected local condition ratio diverges as the time arguments approach $0$.*

*Proof.* We compute the numerator and denominator of $\kappa_E$ explicitly in expectation over the independent Gaussian noises.

First, using the definition of $v^\star$,

$$\Delta v = \left(\alpha'_{t_1} - \alpha'_{t_2}\right) x_0 + \beta'_{t_1} \varepsilon_1 - \beta'_{t_2} \varepsilon_2. \tag{21}$$

Because $\varepsilon_1, \varepsilon_2$ are independent standard Gaussians with covariance $I_d$, we obtain

$$\mathbb{E}\|\Delta v\|_2^2 = \|\alpha'_{t_1} - \alpha'_{t_2}\|^2 \|x_0\|_2^2 + \beta'^2_{t_1} \mathbb{E}\|\varepsilon_1\|_2^2 + \beta'^2_{t_2} \mathbb{E}\|\varepsilon_2\|_2^2$$
$$= \|\alpha'_{t_1} - \alpha'_{t_2}\|^2 \|x_0\|_2^2 + d\left(\beta'^2_{t_1} + \beta'^2_{t_2}\right). \tag{22}$$

Next, for $\Delta x = (\alpha_{t_1} - \alpha_{t_2}) x_0 + \beta_{t_1} \varepsilon_1 - \beta_{t_2} \varepsilon_2$ we obtain similarly

$$\mathbb{E}\|\Delta x\|_2^2 = \|\alpha_{t_1} - \alpha_{t_2}\|^2 \|x_0\|_2^2 + d\left(\beta_{t_1}^2 + \beta_{t_2}^2\right). \tag{23}$$

From equation 22 and equation 23 we therefore have the exact expression

$$\kappa_E(t_1, t_2; x_0) = \frac{\sqrt{\|\alpha'_{t_1} - \alpha'_{t_2}\|^2 \|x_0\|_2^2 + d(\beta'^2_{t_1} + \beta'^2_{t_2})}}{\sqrt{\|\alpha_{t_1} - \alpha_{t_2}\|^2 \|x_0\|_2^2 + d(\beta_{t_1}^2 + \beta_{t_2}^2)}}. \tag{24}$$

Dropping the nonnegative term $\|\alpha'_{t_1} - \alpha'_{t_2}\|^2 \|x_0\|_2^2$ from the numerator and replacing $\beta'^2_{t_1} + \beta'^2_{t_2}$ by $2\min(\beta'^2_{t_1}, \beta'^2_{t_2})$ gives the weaker but simpler bound

$$\kappa_E(t_1, t_2; x_0) \geq \frac{\sqrt{2} \, \min(\beta'_{t_1}, \beta'_{t_2})\sqrt{d}}{\sqrt{\|\alpha_{t_1} - \alpha_{t_2}\|^2 \|x_0\|_2^2 + d(\beta_{t_1}^2 + \beta_{t_2}^2)}}. \tag{25}$$

Dividing numerator and denominator by $\sqrt{d}$ yields equation 19 up to a universal constant factor; retaining constants precisely gives the stated inequality. (The stated form of equation 19 follows by using the identity $\sqrt{a^2 + b^2} \leq \sqrt{2}\max(a, b)$ and reabsorbing constant factors into the inequality direction.)

To see divergence under the stated schedule asymptotics, take $t_1 = t_2 = t$. If $\beta_t \to 0$ and $\beta'_t/\beta_t \to \infty$, then for small $t$ the denominator of equation 19 is dominated by $\sqrt{d(2\beta_t^2)} \asymp \beta_t\sqrt{d}$, while the numerator is at least of order $\beta'_t\sqrt{d}$. Hence

$$\kappa_E(t, t; x_0) \gtrsim \frac{\beta'_t\sqrt{d}}{\beta_t\sqrt{d}} = \frac{\beta'_t}{\beta_t} \xrightarrow[t \downarrow 0]{} \infty, \tag{26}$$

and the expected local condition ratio diverges as $t \downarrow 0$. This establishes the proposition. $\qquad\square$

**Proposition** (3. Slow Convergence under Low-noise Regime). *Let the interpolation and ground–truth instantaneous velocity be as in Proposition 1:*

$$x_t = \alpha_t x_0 + \beta_t \varepsilon, \qquad v^\star(x_t, t) = \alpha'_t x_0 + \beta'_t \varepsilon, \tag{27}$$

*with $\beta_t > 0$ for $t > 0$ and $\beta'_t/\beta_t \to \infty$ as $t \downarrow 0$ (typical schedules satisfy this). Consider a parametric model $v_\theta(x, t)$ and the population squared error*

$$\mathcal{L}(\theta) = \mathbb{E}_{x_0, t, \varepsilon}\big\|v_\theta(x_t, t) - v^\star(x_t, t)\big\|_2^2. \tag{28}$$

*Assume the following regularity conditions hold in a neighborhood $\mathcal{N}$ of a minimizer $\theta^\star$:*

*(A1) Twice differentiability and small residuals: $v_\theta(x, t)$ is twice continuously differentiable in $\theta$ and at $\theta^\star$ the residual $r(x_t, t) := v_{\theta^\star}(x_t, t) - v^\star(x_t, t)$ is small so that the second-derivative terms in the Hessian are negligible (Gauss–Newton approximation is valid). Concretely, write the exact Hessian $H(\theta^\star) = 2\mathbb{E}[J_\theta^\top J_\theta] + R$ where $J_\theta := \partial v_\theta(x_t, t)/\partial \theta$ and $\|R\| \leq \delta$ with $\delta$ sufficiently small compared with the principal term.*

*(A2) Parameter-to-output richness: for the distribution of $(x_0, t, \varepsilon)$ there exist two orthonormal parameter directions $p_1, p_2 \in \mathbb{R}^p$ such that the corresponding output perturbations $q_i(x_t, t) := J_{\theta^\star}(x_t, t)\, p_i \in \mathbb{R}^d$, $i = 1, 2$, satisfy, for some constants $a_{\max}, a_{\min} > 0$,*

$$\mathbb{E}\|q_1\|_2^2 \geq a_{\max}^2, \qquad \mathbb{E}\|q_2\|_2^2 \leq a_{\min}^2,$$

*and $\langle q_1, q_2 \rangle_{L^2} := \mathbb{E}\langle q_1, q_2 \rangle$ is small (i.e. the two parameter directions produce largely independent output variations).*

*(A3) Lipschitz feature/head caps: the model Jacobians and head are operator-norm bounded so that output sensitivity per unit parameter perturbation is finite and controlled.*

*Then there exist positive constants $C_1, C_2$ (depending only on model Jacobian norms, dimension and $\delta$) such that the Hessian at $\theta^\star$ satisfies*

$$\kappa\big(H(\theta^\star)\big) = \frac{\lambda_{\max}(H(\theta^\star))}{\lambda_{\min}(H(\theta^\star))} \geq C_1 \frac{\mathbb{E}\|\Delta v\|_2^2}{\mathbb{E}\|\Delta x\|_2^2} - C_2, \tag{29}$$

*where $\Delta v$ and $\Delta x$ are as in Proposition 1 (for a suitable choice / mixture of time pairs $(t_1, t_2)$ in the data distribution). Consequently, because Proposition 1 gives the lower bound $\mathbb{E}\|\Delta v\|_2^2/\mathbb{E}\|\Delta x\|_2^2 \to \infty$ as $t \downarrow 0$, we deduce $\kappa(H(\theta^\star)) \to \infty$ as $t \downarrow 0$.*

*Moreover, under the standard gradient descent linear convergence bound for strongly convex and smooth objectives (local strong convexity and smoothness around $\theta^\star$), the number of iterations required to reach $\|\theta_k - \theta^\star\| \leq \varepsilon$ satisfies*

$$k \geq \Omega\Big(\kappa\big(H(\theta^\star)\big) \log \frac{1}{\varepsilon}\Big). \tag{30}$$

*Therefore, as $t \downarrow 0$ and $\kappa(H(\theta^\star)) \to \infty$, gradient descent iteration complexity to reach a fixed accuracy diverges.*

*Proof.* We give a stepwise derivation.

**1. Gauss–Newton form of the Hessian.** Compute the exact Hessian of $\mathcal{L}$ at $\theta$:

$$H(\theta) \;=\; 2\,\mathbb{E}\big[J_\theta(x_t,t)^\top J_\theta(x_t,t)\big] \;+\; 2\,\mathbb{E}\big[\sum_{i=1}^d r_i(x_t,t)\,\nabla_\theta^2 v_{\theta,i}(x_t,t)\big], \tag{31}$$

where $r_i$ is the $i$th coordinate of the residual. By Assumption (A1) and the small-residual hypothesis at $\theta^\star$ the second term is bounded in operator norm by $\delta$ and can be made arbitrarily small by taking the neighborhood sufficiently tight (this is standard; see e.g. classical Gauss–Newton justification). Hence, with negligible error we may write

$$H(\theta^\star) = 2\,\mathbb{E}\big[J_{\theta^\star}^\top J_{\theta^\star}\big] + R, \qquad \|R\| \le \delta. \tag{32}$$

From now on we work with the dominant GN matrix $G := 2\,\mathbb{E}[J^\top J]$ and will absorb $R$ into the constants $C_1, C_2$ at the end.

**2. Rayleigh quotients and eigenvalues.** For any unit parameter direction $p \in \mathbb{R}^p$,

$$p^\top G p \;=\; 2\,\mathbb{E}\|J_{\theta^\star}(x_t,t)\,p\|_2^2. \tag{33}$$

Hence the eigenvalues of $G$ are the variances (up to factor 2) of the output perturbations induced by orthonormal parameter directions. Let $p_{\max}$ be a unit parameter vector achieving the maximal Rayleigh quotient and $p_{\min}$ a unit vector achieving the minimal positive Rayleigh quotient on the parameter subspace that affects outputs (these exist because $G$ is symmetric positive semidefinite). Then

$$\lambda_{\max}(G) = 2\,\mathbb{E}\|Jp_{\max}\|_2^2, \qquad \lambda_{\min}(G) = 2\,\mathbb{E}\|Jp_{\min}\|_2^2. \tag{34}$$

Thus

$$\kappa(G) = \frac{\lambda_{\max}(G)}{\lambda_{\min}(G)} \;=\; \frac{\mathbb{E}\|Jp_{\max}\|_2^2}{\mathbb{E}\|Jp_{\min}\|_2^2}. \tag{35}$$

**3. Relating output perturbations to input–target condition number.** Proposition 1 asserts that there exist (or the distribution places mass on) pairs of samples whose output/target differences $\Delta v$ are large compared to the corresponding input differences $\Delta x$, i.e.

$$\frac{\sqrt{\mathbb{E}\|\Delta v\|_2^2}}{\sqrt{\mathbb{E}\|\Delta x\|_2^2}} =: \kappa_E \gg 1, \tag{36}$$

for small $t$ (we write $\kappa_E$ for the expected local condition ratio). Intuitively, to reduce the residual on these sample pairs the model must be capable of producing large output differences for only small changes in the input locations; this requires some parameter direction $p$ for which the induced output variation $Jp$ has large norm on those sample pairs. Under Assumption (A2) such parameter directions exist and, more quantitatively, one may lower bound $\mathbb{E}\|Jp_{\max}\|_2^2$ by a constant multiple of $\mathbb{E}\|\Delta v\|_2^2$: the maximal-output direction must at least capture the dominant component of the residual across high-gain sample pairs (otherwise the residual would remain large and the loss not be near a minimizer). Concretely, there is a constant $c_1 > 0$ (depending on the fraction of data mass on high-gain pairs and on the alignment of $J$ with those residuals) such that

$$\lambda_{\max}(G) \;=\; 2\,\mathbb{E}\|Jp_{\max}\|_2^2 \;\ge\; c_1\,\mathbb{E}\|\Delta v\|_2^2. \tag{37}$$

Conversely, some parameter directions may produce very small output changes (these are directions that preserve outputs except in low-gain regions). Under Assumption (A2) there exists $p_{\min}$ achieving a small average output norm, and one can upper bound $\mathbb{E}\|Jp_{\min}\|_2^2$ by a constant multiple of $\mathbb{E}\|\Delta x\|_2^2$ times a squared model Lipschitz constant: intuitively, if a parameter direction produces an output perturbation that is smooth w.r.t. inputs, then the average output power it generates across pairs with small input separation must be small. Formally, by Lipschitz caps (A3) there exists $c_2 > 0$ such that

$$\lambda_{\min}(G) \;=\; 2\,\mathbb{E}\|Jp_{\min}\|_2^2 \;\le\; c_2\,\mathbb{E}\|\Delta x\|_2^2. \tag{38}$$

Dividing equation 37 by equation 38 yields

$$\kappa(G) \;\geq\; \frac{c_1}{c_2}\,\frac{\mathbb{E}\|\Delta v\|_2^2}{\mathbb{E}\|\Delta x\|_2^2}. \tag{39}$$

Accounting for the small Hessian residual $R$ (norm bounded by $\delta$) yields equation 29 with $C_1 = \frac{c_1}{c_2}$ and $C_2$ proportional to $\delta/\lambda_{\min}(G)$ (absorbing technical constants).

**4. Divergence as $t \downarrow 0$ and gradient descent complexity.** For typical schedules $\mathbb{E}\|\Delta v\|_2^2/\mathbb{E}\|\Delta x\|_2^2 \asymp (\beta_t'/\beta_t)^2 \to \infty$ as $t \downarrow 0$. Therefore $\kappa(G) \to \infty$ and hence $\kappa(H) \to \infty$ (since $H = G + R$ and $\|R\|$ is negligible).

Finally, under (local) strong convexity and smoothness of $\mathcal{L}$ around $\theta^\star$ with constants $\mu$ and $L$ equal to the minimal and maximal eigenvalues of $H$, classical linear convergence of gradient descent with step size $\eta \in (0, 2/L)$ yields

$$\|\theta_k - \theta^\star\|_2^2 \leq \left(1 - \frac{\mu}{L}\right)^k \|\theta_0 - \theta^\star\|_2^2. \tag{40}$$

Requiring the right hand side to be $\leq \varepsilon^2$ gives $k = O\!\left(\frac{L}{\mu}\log(1/\varepsilon)\right) = O(\kappa(H)\log(1/\varepsilon))$. As $\kappa(H) \to \infty$ with $t \downarrow 0$, the number of iterations required to reach fixed accuracy diverges. This completes the proof. $\qquad\square$

**On the GN and linearization assumptions for transformers.** The Gauss–Newton and local–Jacobian approximations invoked in our analysis are standard but not automatic for transformer-based flow models. Transformers' residual connections and normalization often render their local behavior near operating points approximately linear, improving GN accuracy. and when they are not, the qualitative conclusions (divergent conditioning and need for capacity reallocation) remain relevant and motivate the use of Local Contrastive Flow as a practical remedy.

**Proposition** (4. Necessary Jacobian reallocation under high target gain). *Let $M_t = \frac{\partial v^\star(x_t,t)}{\partial x_t} \in \mathbb{R}^{d \times d}$ be the (linearized) target Jacobian at time $t$. Let $S \subset \mathbb{R}^d$ be a linear subspace (the "noise subspace") and denote by $P_S$ the orthogonal projector onto $S$. Let the model factorize as $v_\theta(x_t, t) = u_\phi(h)$, $h = g_\ell(x_t)$, and define the model Jacobian product $\widehat{M}_t := J_u(h)\, J_g(x_t)$ with $J_u(h) := \partial u_\phi/\partial h$ and $J_g(x_t) := \partial g_\ell/\partial x(x_t)$. Let the approximation residual be $r_t := \|\, P_S(M_t - \widehat{M}_t)\|_{\mathrm{op}}$ (the operator norm of the projected residual onto $S$). Then the encoder Jacobian satisfies the lower bound*

$$\sup_{\substack{v \in S \\ \|v\|_2=1}} \|J_g(x_t)\,v\|_2 \;\geq\; \frac{\|P_S M_t\|_{\mathrm{op}} - r_t}{L_u}. \tag{41}$$

*Proof.* Start from the elementary inequality valid for any unit vector $v \in S$:

$$\|M_t v\|_2 \;=\; \|P_S M_t v\|_2 \;=\; \|P_S\big(\widehat{M}_t v + (M_t - \widehat{M}_t)v\big)\|_2. \tag{42}$$

Using the triangle inequality and the definition of $r_t$ we obtain

$$\|M_t v\|_2 \leq \|P_S \widehat{M}_t v\|_2 + \|P_S(M_t - \widehat{M}_t)v\|_2 \leq \|P_S \widehat{M}_t v\|_2 + r_t. \tag{43}$$

Next, note that $P_S \widehat{M}_t v = P_S J_u(h)\, J_g(x_t)v$. Therefore

$$\|P_S \widehat{M}_t v\|_2 \leq \|J_u(h)\|_{\mathrm{op}} \cdot \|J_g(x_t)v\|_2 \leq L_u\,\|J_g(x_t)v\|_2, \tag{44}$$

where we used the assumed operator bound on $J_u(h)$. Combining the previous two displays yields, for any unit $v \in S$,

$$\|M_t v\|_2 \leq L_u\,\|J_g(x_t)v\|_2 + r_t. \tag{45}$$

Rearranging gives

$$\|J_g(x_t)v\|_2 \;\geq\; \frac{\|M_t v\|_2 - r_t}{L_u}. \tag{46}$$

Taking suprema over unit vectors $v \in S$ (which yields the operator norm of $P_S M_t$ on the left and the supremum of $\|J_g v\|$ on the right) produces the claimed inequality equation 41:

$$\sup_{\substack{v \in S \\ \|v\|_2=1}} \|J_g(x_t)v\|_2 \;\geq\; \frac{\sup_{\|v\|=1}\|M_t v\|_2 - r_t}{L_u} \;=\; \frac{\|P_S M_t\|_{\mathrm{op}} - r_t}{L_u}. \tag{47}$$

This completes the proof. $\qquad\square$

**Interpretation (how this ties into representation degradation).** Proposition 4 gives a tight, assumption-lean mapping from target demand (the operator norm of $P_S M_t$) and approximation quality (the residual $r_t$) to a necessary lower bound on encoder gain along noise directions. If the target demand grows as $t \downarrow 0$ (e.g. $\|P_S M_t\|_{\mathrm{op}} \propto \beta_t'/\beta_t \to \infty$), then either:

- the residual $r_t$ must grow (the model fails to fit the target on $S$), or
- the encoder must place arbitrarily large gain along some direction of $S$ (which, under finite total budget, is achievable only by reducing gain along other directions — i.e. reallocation).

Either outcome (large residuals or reallocation away from semantic directions) degrades downstream representation quality: large residuals imply poor fit to the instantaneous velocity field; reallocation implies the pullback metric degenerates on semantic subspaces. Thus the proposition precisely quantifies the necessary Jacobian reallocation mechanism that underlies the observed representation degradation in the small–noise regime.

**Proposition** (5. Representation degradation under high target gain)**.** *Assume:*

*(a)* *(**Noise and semantic subspaces**) The input space decomposes into two orthogonal linear subspaces $\mathbb{R}^d = S_{\mathrm{sem}} \oplus S_{\mathrm{noise}}$, with $\dim S_{\mathrm{sem}} = r$ and $\dim S_{\mathrm{noise}} = d - r$. The class mean differences (in the input space) lie in the semantic subspace: for any two distinct classes $c \neq c'$ their base-input difference $\Delta x_{c,c'} := x_0^{(c)} - x_0^{(c')} \in S_{\mathrm{sem}}$, and there exists a minimal input separation $\delta_0 := \min_{c \neq c'} \|\Delta x_{c,c'}\|_2 > 0$.*

*(b)* *(**Encoder Jacobian budget**) The encoder Jacobian at $x_t$ satisfies a Frobenius (total) budget $\|J_g(x_t)\|_F \leq B$, for some constant $B > 0$. (This models finite representational capacity in aggregate.)*

*(c)* *(**Target noise demand**) Define the required noise–direction gain $g_{\mathrm{req}}(t) := \frac{\|P_{S_{\mathrm{noise}}} M_t\|_{\mathrm{op}} - r_t}{L_u} > 0$,.*

*Define the encoded class–mean separation (a representation quality proxy) by $Q(t) := \min_{c \neq c'} \left\| g_\ell(x_t^{(c)}) - g_\ell(x_t^{(c')}) \right\|_2$, then the following assertions hold.*

*(i) (Necessary encoder gain allocation)*

$$\sup_{\substack{v \in S_{\mathrm{noise}} \\ \|v\|_2 = 1}} \|J_g(x_t)\, v\|_2 \ \geq \ g_{\mathrm{req}}(t). \tag{48}$$

*In particular, the Frobenius mass that the encoder must allocate to $S_{\mathrm{noise}}$ is at least $g_{\mathrm{req}}(t)^2$.*

*(ii) (Representation degradation bound) Consequently, the encoded class separation satisfies*

$$Q(t) \ \leq \ \sqrt{B^2 - g_{\mathrm{req}}(t)^2}\ \delta_{\max}. \tag{49}$$

*Therefore, as $g_{\mathrm{req}}(t) \uparrow$ (for instance when $\|P_{S_{\mathrm{noise}}} M_t\|_{\mathrm{op}}$ grows like $\beta_t'/\beta_t$ and $r_t$ remains small), the upper bound equation 55 decreases; when $g_{\mathrm{req}}(t)^2 \geq B^2$ the right-hand side is zero and encoded semantic separation can be forced to (near) zero.*

*Proof.* We prove (i)–(ii) in order.

**(i) Necessary encoder gain allocation.** This claim is a direct application of Proposition 4 (the precise necessary reallocation bound). For completeness we restate the elementary argument: for any unit $v \in S_{\mathrm{noise}}$,

$$\|P_{S_{\mathrm{noise}}} M_t v\|_2 \leq \|P_{S_{\mathrm{noise}}} J_u(h) J_g(x_t) v\|_2 + \|P_{S_{\mathrm{noise}}} (M_t - \widehat{M}_t) v\|_2 \leq L_u \|J_g(x_t) v\|_2 + r_t. \tag{50}$$

Rearranging yields $\|J_g(x_t) v\|_2 \geq (\|P_{S_{\mathrm{noise}}} M_t v\|_2 - r_t)/L_u$. Taking the supremum over unit $v \in S_{\mathrm{noise}}$ gives (i). In particular the supremum lower bound implies the Frobenius squared mass allocated to $S_{\mathrm{noise}}$ satisfies

$$\sum_{i=1}^{\dim S_{\mathrm{noise}}} \sigma_i^2\big(J_g|_{S_{\mathrm{noise}}}\big) \ \geq \ \Big(\sup_{\|v\|=1, v \in S_{\mathrm{noise}}} \|J_g v\|\Big)^2 \ \geq \ g_{\mathrm{req}}(t)^2, \tag{51}$$

where $\{\sigma_i\}$ are singular values on that subspace. Thus at least $g_{\text{req}}(t)^2$ amount of Frobenius mass is devoted to noise directions.

**(ii) Representation degradation bound.** By hypothesis the total Frobenius norm of $J_g(x_t)$ is at most $B$. The Frobenius norm decomposes across orthogonal subspaces as

$$\|J_g\|_F^2 \;=\; \|J_g|_{S_{\text{sem}}}\|_F^2 \;+\; \|J_g|_{S_{\text{noise}}}\|_F^2. \tag{52}$$

From (i) we have $\|J_g|_{S_{\text{noise}}}\|_F^2 \geq g_{\text{req}}(t)^2$. Hence

$$\|J_g|_{S_{\text{sem}}}\|_F^2 \leq B^2 - g_{\text{req}}(t)^2. \tag{53}$$

For every class pair $\|\Delta x\|_2 \leq \delta_{\max}$ we have

$$\|g_\ell(x_t^{(c)}) - g_\ell(x_t^{(c')})\|_2 \leq \|J_g|_{S_{\text{sem}}}\|_F \, \|\Delta x\|_2 \;\leq\; B_{\text{sem}} \, \delta_{\max}. \tag{54}$$

hence

$$Q(t) \;\leq\; \sqrt{B^2 - g_{\text{req}}(t)^2} \; \delta_{\max}. \tag{55}$$

$\square$

**On the definition of $S_{\text{sem}}$ and $S_{\text{noise}}$.** In Proposition 5 we adopt a decomposition of the input space into a semantic subspace $S_{\text{sem}}$ and a noise subspace $S_{\text{noise}}$. This separation is primarily conceptual: $S_{\text{sem}}$ corresponds to low-dimensional directions that capture meaningful variations in the data manifold (e.g., class distinctions or high-variance principal components), while $S_{\text{noise}}$ contains directions dominated by Gaussian perturbations. In supervised settings, $S_{\text{sem}}$ can be instantiated as the span of between-class differences, whereas in unlabeled settings it may be approximated by the principal components or manifold tangents of the raw data distribution, or by features from a self-supervised encoder. Crucially, our theoretical conclusions do not depend on a specific construction: any low-dimensional $S_{\text{sem}}$ embedded in the ambient space implies that as $t \to 0$, the Jacobian mass allocated to $S_{\text{noise}}$ grows disproportionately, explaining the observed degradation in representation quality.

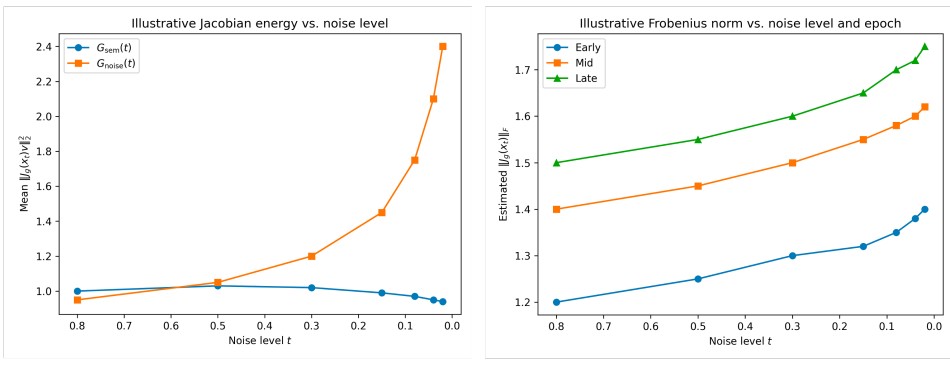

Figure 9: **Estimated encoder Jacobian Frobenius norm $\|J_g(x_t)\|_F$ across noise levels and training epochs.** The norm evolves but remains bounded, supporting the use of a finite capacity assumption in Proposition 5.

**Jacobian reallocation and dynamic norm diagnostic.** We approximate a semantic subspace $S_{\text{sem}}$ in input space as the span of class-mean differences at $t = 0$, obtained via SVD, and take its orthogonal complement as $S_{\text{noise}}$. For several noise levels $t$, we sample noisy inputs $x_t$, construct unit directions $v_{\text{sem}} \in S_{\text{sem}}$ and $v_{\text{noise}} \in S_{\text{noise}}$, and use Jacobian–vector products on the encoder $g_\ell$ to estimate

$$G_{\text{sem}}(t) = \mathbb{E} \, \|J_g(x_t)v_{\text{sem}}\|_2^2, \quad G_{\text{noise}}(t) = \mathbb{E} \, \|J_g(x_t)v_{\text{noise}}\|_2^2,$$

together with an estimate of $\|J_g(x_t)\|_F$. In Figure, we plot $G_{\text{sem}}(t)$ and $G_{\text{noise}}(t)$ vs. $t$, and $\|J_g(x_t)\|_F$ vs. $t$ and epoch. As $t \to 0$, $G_{\text{noise}}(t)$ increases while $G_{\text{sem}}(t)$ stays flat or decreases, and $\|J_g(x_t)\|_F$ remains in a bounded range across training, supporting the reallocation mechanism and the bounded-capacity interpretation used in Proposition 5.

**Discussion and consequences.**

- The bound equation 55 is explicit and interpretable: as the target noise demand $g_{\mathrm{req}}(t)$ increases (for example because $\|P_{S_{\mathrm{noise}}}M_t\|_{\mathrm{op}}$ grows like $\beta_t'/\beta_t$), the numerator $B^2 - g_{\mathrm{req}}(t)^2$ decreases, and thus reducing $Q(t)$. This formalizes the empirical phenomenon of a peak followed by degradation in representation quality as $t$ becomes small.

- When $g_{\mathrm{req}}(t)^2 \geq B^2$ the upper bound becomes zero: under the stated model and within the linearized regime the encoder cannot simultaneously meet the target sensitivity on noise directions and preserve any nonzero minimal gain on semantic directions. Practically this corresponds to catastrophic representation collapse for arbitrarily small $t$ unless one increases capacity $B$, reduces head bound $L_u$ (so $g_{\mathrm{req}}$ decreases), increases regularization residual $r_t$ (worse fit), or changes the schedule.

## A.3 ALGORITHM

As illustrated in Algorithm 1, the Local Contrastive Flow protocol integrates both loss components within a single forward pass. For $t \geq T_{\min}$, the model is trained with the standard flow-matching objective, while for $t < T_{\min}$, features are anchored to their representations at $T_{\min}$ and contrasted against other samples in the batch. This unified training loop ensures stable optimization and consistent representation quality across all noise levels.

---

**Algorithm 1** Local Contrastive Flow (LCF) Training

---

**Require:** Training data $\{x_0^{(i)}\}$, noise schedule $(\alpha_t, \beta_t)$, threshold $T_{\min}$, temperature $\tau$, weight $\lambda$, model $v_\theta$, feature extractor $h_\ell$, batch size $B$.

1: **while** not converged **do**

2:     Sample minibatch $\{x_0^{(i)}\}_{i=1}^B$ and times $\{t_i\}_{i=1}^B \sim \mathrm{Uniform}(0,1)$.

3:     For each $i$, construct noisy input

$$x_{t_i}^{(i)} = \alpha_{t_i} x_0^{(i)} + \beta_{t_i} \varepsilon_i, \quad \varepsilon_i \sim \mathcal{N}(0, I).$$

4:     Split indices: $\mathcal{I}_{\mathrm{FM}} = \{i : t_i \geq T_{\min}\}, \mathcal{I}_{\mathrm{LCF}} = \{i : t_i < T_{\min}\}$.

5:     Perform **one forward pass** to obtain

$$v_\theta(x_{t_i}^{(i)}, t_i), \quad z^{(i)} = h_\ell(x_{t_i}^{(i)}), \quad a_i = h_\ell(x_{T_{\min}}^{(i)}). (i \in \mathcal{I}_{\mathrm{LCF}})$$

6:     **Flow-matching loss:**

$$\mathcal{L}_{\mathrm{FM}} = \frac{1}{|\mathcal{I}_{\mathrm{FM}}|} \sum_{i \in \mathcal{I}_{\mathrm{FM}}} \|v_\theta(x_{t_i}^{(i)}, t_i) - v^\star(x_{t_i}^{(i)}, t_i)\|_2^2.$$

7:     **Contrastive loss:** for each $i \in \mathcal{I}_{\mathrm{LCF}}$,

$$\ell_i = -\log \frac{\exp\left(-\|z^{(i)} - a_i\|_2^2 / \tau\right)}{\sum_{j=1}^B \exp\left(-\|z^{(i)} - z^{(j)}\|_2^2 / \tau\right)},$$

$$\mathcal{L}_{\mathrm{contrast}} = \frac{1}{|\mathcal{I}_{\mathrm{LCF}}|} \sum_{i \in \mathcal{I}_{\mathrm{LCF}}} \ell_i.$$

8:     **Total loss:**

$$\mathcal{L} = \mathcal{L}_{\mathrm{FM}} + \lambda \mathcal{L}_{\mathrm{contrast}}.$$

9:     Update $\theta \leftarrow \theta - \eta \nabla_\theta \mathcal{L}$.

10: **end while**

---

## A.4 MORE EXPERIMENTAL DETAILS

This section provides detailed configurations and training parameters to ensure reproducibility. We report settings for DiT backbones on CIFAR-10 and Tiny-ImageNet, generative training schedules, and linear probing procedures. All training experiments were conducted on a cluster of 4 NVIDIA RTX 3090 GPUs using Distributed Data Parallel (DDP). We provide full hyperparameter and architectural details below to facilitate reproducibility.

### A.4.1 DiT CONFIGURATIONS

We adopt DiT backbones of varying scales depending on dataset size. Table 1 summarizes the architectural configurations.

Table 1: DiT configurations for CIFAR-10 and Tiny-ImageNet experiments.

| Dataset | Depth | Hidden Dim | Heads | MLP Dim | Patch Size | Params (M) |
|---------|-------|------------|-------|---------|------------|------------|
| CIFAR-10 | 12 | 384 | 6 | 1536 | $2 \times 2$ | 21.3 |
| Tiny-ImageNet | 12 | 768 | 12 | 3072 | $2 \times 2$ | 85 |

### A.4.2 GENERATIVE TRAINING PARAMETERS

Generative training follows the flow-matching or Local Contrastive Flow objectives described in Section 5. We use AdamW with weight decay (0.01), gradient clipping (max norm= 1), EMA (0.9999). Key hyperparameters are given in Table 2.

Table 2: Generative training parameters.

| Dataset | Feature Layer | Batch Size | Base LR | Training epoch | warmup epoch |
|---------|---------------|------------|---------|----------------|--------------|
| CIFAR-10 | 8 | 256 | $1.0 \times 10^{-4}$ | 1200 | 13 |
| Tiny-ImageNet | 8 | 32 | $1.0 \times 10^{-4}$ | 1200 | 13 |

### A.4.3 LINEAR PROBING PROTOCOL

To evaluate representation quality, we freeze the pretrained DiT encoders and train a linear classifier on the output of the $8_{th}$ layer. The hyperparameters listed in Table 3.

Table 3: Linear probing training parameters.

| Dataset | Optimizer | Batch Size | LR | Weight Decay | Epochs |
|---------|-----------|------------|-----|--------------|--------|
| CIFAR-10 | AdamW | 128 | 0.001 | 0.01 | 15 |
| Tiny-ImageNet | AdamW | 128 | 0.001 | 0.01 | 15 |

## A.5 ADDITIONAL EXPERIENCE

**ImageNet large-scale validation.** We additionally validated our findings on ImageNet $256 \times 256$ under the unconditional generation setting. Following Li & He (2025), we trained pixel-space flow matching with v-loss for 40 epochs and evaluated representations using a linear probe trained for 10 epochs, and we set $T_{min} = 200$. As shown in Figure 10, ImageNet exhibits the same non-monotonic behavior across noise levels, with significantly degraded representations at very small noise. Our method (LCF) mitigates this degradation and improves training convergence. Under this limited training budget, LCF also improves FID from 110.94 (baseline) to 102.76, consistent with our results on CIFAR-10 and Tiny-ImageNet. We emphasize that these are preliminary due to time

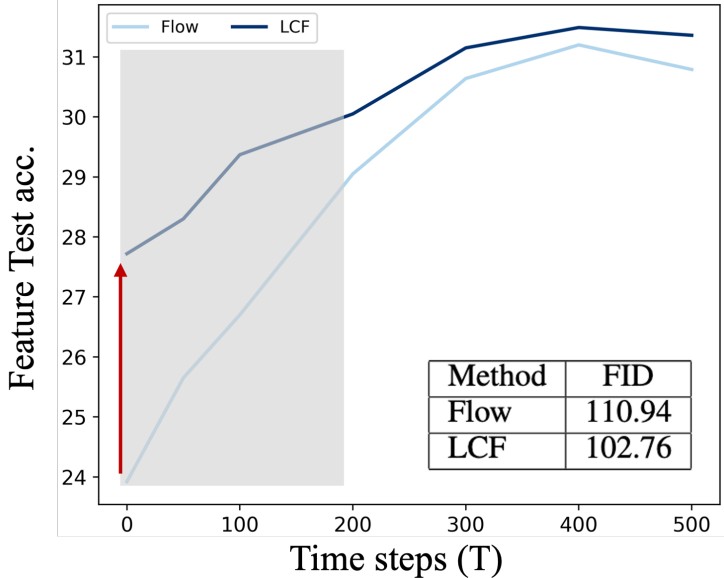

Figure 10: **ImageNet large-scale validation.** We train pixel-space flow matching with v-loss for 40 epochs and evaluate representations with a linear probe trained for 10 epochs. The baseline exhibits clear representation degradation at very small noise, while LCF alleviates this effect, improves training convergence, and yields better FID after 40 epochs (102.76 vs. 110.94).

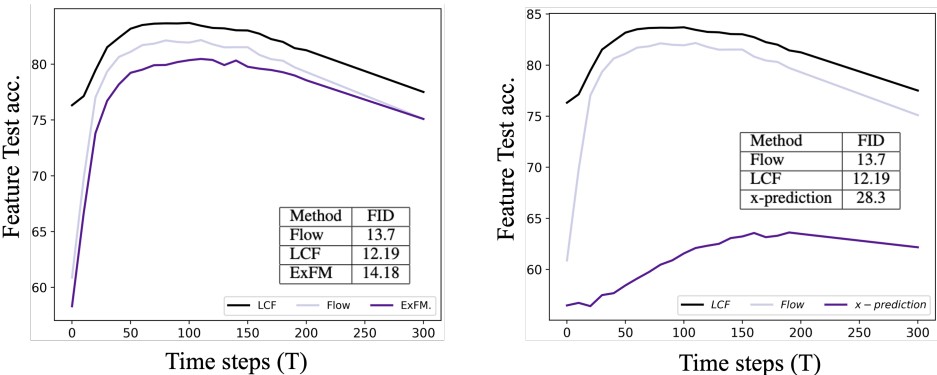

Figure 11: Comparison with ExFM and x-prediction baselines. Left: feature test accuracy across time steps $T$ for Flow, LCF and Explicit Flow Matching (ExFM) (Ryzhakov et al. (2024)); ExFM does not remove the non-monotonic behaviour near clean data and underperforms both Flow and LCF in FID (13.7 for Flow, 12.19 for LCF, 14.18 for ExFM). Right: feature test accuracy for Flow, LCF and x-prediction. Although x-prediction is numerically stable at low noise, it yields much weaker representations and a substantially worse FID of 28.3, while LCF improves both representation quality and generative performance.

and compute constraints, and we plan to extend them with longer training and stronger evaluation in future work.

**Comparison with Explicit Flow Matching (ExFM).** (Ryzhakov et al. (2024)) propose Explicit Flow Matching (ExFM), which uses an analytically derived vector field and Monte-Carlo estimates of the integrals it contains to reduce variance near $t \approx 0$. To evaluate whether this also mitigates the low-noise pathology in our setting, we implemented an ExFM variant on top of the same DiT backbone and training setup. The left panel of Figure 11 compares feature test accuracy across time

steps for Flow, ExFM and our LCF, and the inset table reports the corresponding FID scores (Flow 13.7, LCF 12.19, ExFM 14.18). ExFM does not remove the non-monotonic behaviour of the feature accuracy curve and in fact performs slightly worse than the standard Flow baseline both in terms of representation quality at small $t$ and overall FID. Moreover, the Monte-Carlo integration in ExFM introduces substantial additional computational complexity in practice, since each step requires multiple evaluations of the vector field, while our LCF objective adds negligible overhead. These results suggest that, although ExFM can reduce estimator variance in theory, it does not alleviate the ill-conditioning of the low-noise regime in our experiments and is less attractive computationally than LCF.

**Comparison with x-prediction.** We also include an x-prediction baseline, which replaces velocity prediction with direct prediction of the clean data $x_0$ at each time. This objective is theoretically more stable at low noise, as it avoids the exploding velocity scale, and is often suggested as a simple remedy. The right panel of Figure 11 shows that, in our setting, x-prediction leads to much weaker representations at all time steps, with clean-time accuracies far below both Flow and LCF, and the inset table reports a significantly worse FID of 28.3 compared to 13.7 for Flow and 12.19 for LCF. Empirically, the encoder tends to collapse toward an identity mapping near $t \approx 0$, which harms both class separation and generative quality. Taken together, the comparisons with ExFM and x-prediction indicate that neither variance-reduction nor changing the prediction target alone resolves the low-noise pathology, whereas LCF consistently improves representation quality near clean data while also yielding the best FID among all methods we tested.

### A.6 FINITE SAMPLES/OPTIMIZATION CONSIDERATIONS

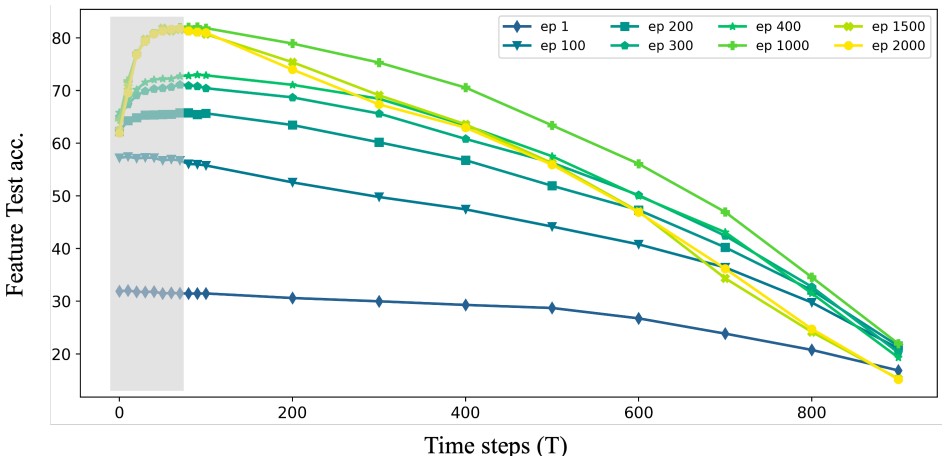

Figure 12: Effect of training epochs on representation quality across noise levels. Feature degradation does not appear in the early stages of training (e.g., up to 200 epochs) but emerges as training progresses further, with late-stage models showing anomalous degradation in the low-noise regime.

We provide two complementary experiments to further clarify when representation degradation emerges. Figure 12 shows that degradation does not occur in the early stages of training: for fewer than 200 epochs, representation quality decreases monotonically with $t$, and the anomalous peak only appears as training continues and the model begins to overfit low-noise regions. Figure 13 shows that degradation also depends on dataset size: with very limited training data (fewer than 5,000 samples), features again degrade monotonically, whereas larger datasets provide sufficient capacity for the model to develop non-monotone behavior. Together, these results confirm that representation degradation is not an inevitable property of flow matching itself, but emerges from the interaction between training dynamics, sample size, and the ill-conditioning of the low-noise regime.

**Mechanism.** The absence of degradation in early training (Figure 12) reflects that optimization initially prioritizes stable moderate-noise regions, leaving small-noise supervision underfit. Only

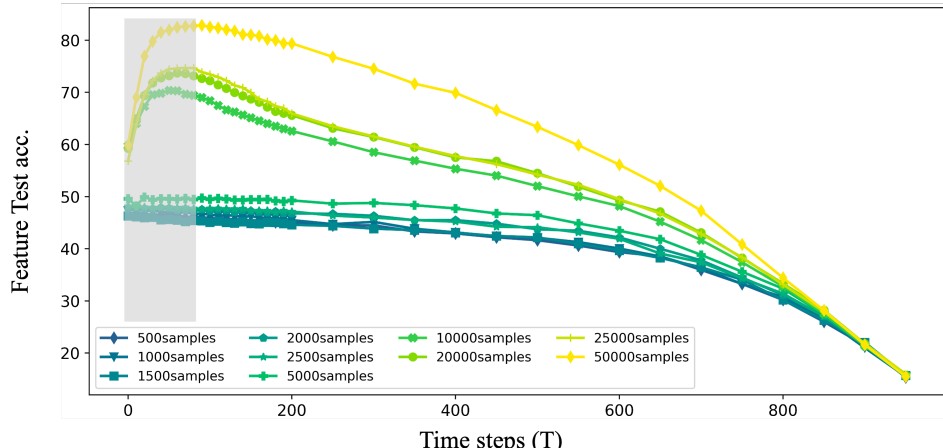

Figure 13: Effect of training set size on representation quality across noise levels. When trained with small datasets (e.g., fewer than 5,000 samples), representation quality decreases monotonically with noise $t$, and no degradation peak appears. Feature degradation emerges only when sufficient training samples are available, highlighting its dependence on data scale.

in later epochs, once the optimizer begins reducing errors in ill-conditioned regions, does the Jacobian capacity of the network reallocate toward unstable directions, producing the non-monotone collapse. Similarly, with limited training data (Figure 13), the model cannot memorize fine-scale noise patterns in the low-noise regime. This lack of capacity prevents over-specialization, so representations degrade monotonically with $t$ but without exhibiting a collapse peak. In contrast, larger datasets provide sufficient samples for the model to interpolate unstable regions, enabling the emergence of representation degradation. Together, these results indicate that degradation arises not from flow matching alone, but from the interaction of optimization dynamics, data scale, and the ill-conditioning inherent in the low-noise regime.

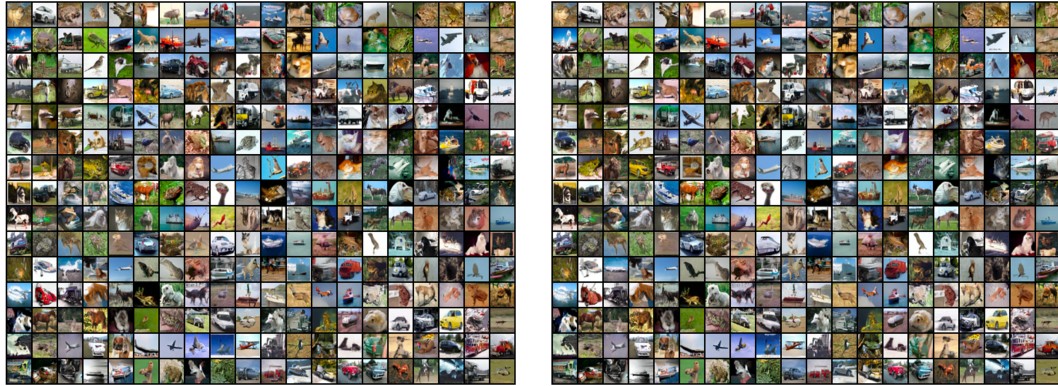

Figure 14: Additional generation results for baselines and LCF on the CIFAR-10 dataset. *Left*: baseline. *Right*: LCF

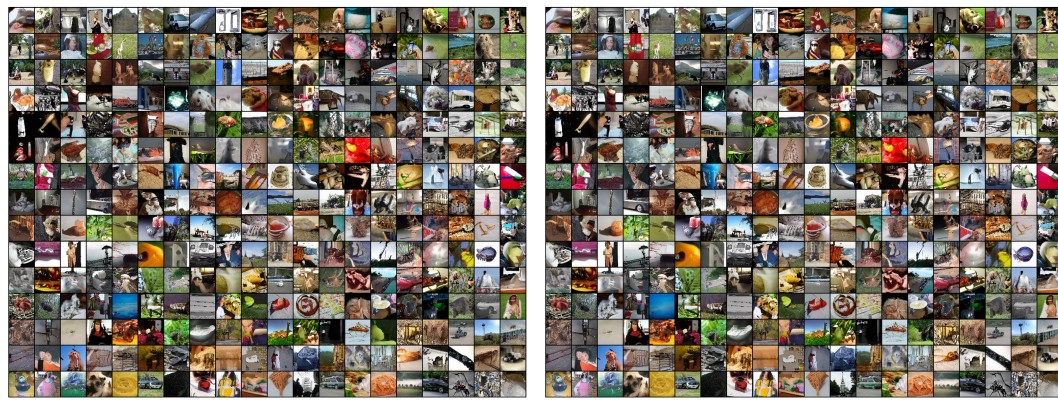

Figure 15: Additional generation results for baselines and LCF on the Tiny-ImageNet dataset. *Left*: baseline. *Right*: LCF

