# OpenReview forum: "Flow Matching in the Low-Noise Regime: Pathologies and a Contrastive Remedy"
_ICLR.cc/2026/Conference — Submitted to ICLR 2026_

### Official Review · Reviewer_ZMUZ · 2025-10-26

**Soundness:** 2
**Presentation:** 2
**Contribution:** 3
**Rating:** 4
**Confidence:** 3

**Summary:**

The paper claims flow matching models break down in the low-noise regime, where small input changes cause large, unstable target velocity shifts, resulting in ill-conditioned training and degraded representations. The authors prove this and show empirical failures on CIFAR-10 and Tiny-ImageNet with DiT models. They propose a hybrid fix: for moderate/high noise ($t \geq T_\text{min}$), use standard flow matching; for low noise ($t < T_\text{min}$), switch to contrastive learning that aligns low-noise features with stable moderate-noise anchors and repels negatives. The paper positions the refined flow matching as a tool for better representation learning in generative models and ties to a much broader scope like large multimodal foundation models and unified models, arguing that generative supervision should balance synthesis and understanding, but the "low-noise issues" disrupt this.

**Strengths:**

1. Identification of Low-Noise Pathology: The paper clearly identifies and formalizes critical instability in flow matching at low noise levels, linking it to divergent condition numbers and representation degradation, addressing an underexplored issue with direct relevance to generative and representation learning.
2. Theoretical Analysis: Provides strong theoretical foundations, including proofs of condition number divergence, slow convergence, and Jacobian reallocation, offering a principled explanation that extends beyond empirical observations.
3. Practical Remedy: Introduces Local Contrastive Flow, a simple hybrid objective that mitigates pathologies, with empirical results on CIFAR-10 and Tiny-ImageNet showing faster convergence, improved FID, and stabilized representations.

**Weaknesses:**

1. Evaluations are confined to small-scale datasets (CIFAR-10, Tiny-ImageNet), lacking experiments on high-resolution (e.g., ImageNet 256x256) or multimodal benchmarks, which undermines claims of broad applicability (unifying generation and understanding) and ignores scalability concerns in real-world generative modeling.
2. The theoretical analysis hinges on idealized conditions like bounded Jacobians, potentially failing to reflect the nonlinear dynamics of transformer-based models in practice.

**Questions:**

You claim low-noise pathology is an "inherent consequence of the generative modeling objective," but your analysis depends on FM's interpolation ($ x_t = \alpha_t x_0 + \beta_t \epsilon $). AR models and most MLLMs (e.g., Janus[1], NTP-based) don't use explicit noise schedules. How does this pathology apply to AR settings with no ($ t \to 0 $) regime? Your Background draws parallels to AR models but provides no concrete evidence or analogy. Isn't this overreach? Please provide a concrete example of this issue arising in a popular MLLM, or clarify that it's FM-specific. Otherwise, why should the MLLM community focus on unified generation and understanding care?

Proposition 5 assumes the input space decomposes cleanly into semantic and noise subspaces with orthogonal projectors, leading to the Frobenius norm decomposition in Equation (51):  $ |J_g|F^2 = |J_g|{S_{\mathrm{sem}}}|F^2 + |J_g|{S_{\mathrm{noise}}}|_F^2 $. However, in high-dimensional vision models like DiT, this orthogonality seems neither unique nor stable during training. Could you provide empirical validation (e.g., via PCA or SVD on learned Jacobians, or measurements of subspace angles) to show that this assumption holds even approximately? If not, how might cross-term interactions (ignored in the decomposition) affect the bound's accuracy in practice, and would incorporating them change your conclusions?

Proposition 5 also assumes a fixed Frobenius norm constraint on the encoder Jacobian ( $ |J_g(x_t)|_F \leq B $ ), independent of noise level t and training progress. Yet, in transformers like DiT, the Jacobian norm evolves dynamically across t and epochs (as hinted in your Figure 8). How does this fixed B relate to actual network capacity, and does the bound remain tight or meaningful throughout training? Could you provide an analysis (e.g., tracking B's effective value as an estimation over epochs or noise schedules in experiments) to show how it holds up, or discuss adaptations for dynamic norms? If the assumption doesn't fully capture real training dynamics, how might relaxing it impact the pathology's interpretation?

[1] C. Wu et al., "Janus: Decoupling Visual Encoding for Unified Multimodal Understanding and Generation," 2024.

---

> ### Author Response · Authors · 2025-11-26
> **# (1/2) Response to Reviewer ZMUZ**
>
> Thank you for your comments and please see our response to the feedbacks below.
>
> ## Response to W1 (large scale evaluation):
> Please refer to General Response (**Regarding scale**).
>
> ## Response to W2 (idealized conditions like bounded Jacobians):
> We agree that our theoretical analysis relies on idealized assumptions such as bounded Jacobians, and that real transformer architectures may violate these assumptions globally. However, **our goal is not to fully characterize the exact nonlinear dynamics of transformers, but to analyze the local behavior in the training regime where flow-matching models actually operate.** In practice, components such as residual connections, normalization layers, weight decay, and gradient clipping implicitly enforce effective Jacobian bounds, making the Gauss–Newton–style analysis a standard and widely used approximation for interpreting optimization behavior in large transformer models [1,2,3].
>
> Moreover, our theoretical results are supported by consistent empirical observations: the predicted condition-number divergence, convergence slowdown, and representation degradation all appear robustly across architectures and datasets. This indicates that the theoretical mechanism we identify is not an artifact of the simplifying assumptions but reflects a structural phenomenon of the flow-matching objective itself. Thus, while the assumptions are idealized, they serve as a practical and insightful lens for understanding the pathology observed in real models.
>
> ### References
> [1] Qin Z, Zhou J, Zhu Z. On the Convergence of Gradient Descent on Learning Transformers with Residual Connections[J]. arXiv preprint arXiv:2506.05249, 2025.
> [2] Qi X, Wang J, Zhang L. Understanding optimization of deep learning via jacobian matrix and lipschitz constant[J]. arXiv preprint arXiv:2306.09338, 2023.
> [3] Zhang B, Jin J, Fang C, et al. Improved analysis of clipping algorithms for non-convex optimization[J]. Advances in Neural Information Processing Systems, 2020, 33: 15511-15521.
>
> ## Response to Q1 (overreached of low-noise pathology):
> We agree that our earlier wording was unclear: the low-noise pathology we identify is specific to flow-matching style generative models, where the interpolation $x_t$ and the $t \to 0$ regime make the objective inherently ill-posed. Autoregressive MLLMs such as Janus or NTP-based models do not use noise schedules, and we do not claim that they exhibit the same behavior. **Our discussion of MLLMs was meant only to motivate why stable generative supervision remains important for the community.**
>
> As multimodal systems increasingly seek unified generative–discriminative visual backbones, models that can jointly support generation, dense prediction, editing, and perception are becoming more valuable. Flow matching is a promising candidate for such unified vision components, which makes understanding and addressing its low-noise instability relevant even if the pathology is FM-specific.

---

> ### Author Response · Authors · 2025-11-26
> **# (2/2) Response to Reviewer ZMUZ**
>
> ## Response to Q2&Q3 (Proposition 5 assumptions):
> We agree that both assumptions in Proposition 5 are idealized. First, the decomposition into semantic/noise subspaces with orthogonal projectors is not meant to imply that **DiTs maintain a unique, fixed semantic subspace throughout training**. Mathematically, for any fixed time we can define $S_{\text{sem}}$ as the span of class-mean differences (or similar discriminative directions) and take its orthogonal complement as $S_{\text{noise}}$; for this **chosen** orthogonal decomposition the Frobenius identity in Eq. (51) holds exactly, without cross terms.
>
> What is idealized is that class-relevant variation is largely concentrated in such a low-dimensional span and that low-noise supervision pushes strongly in directions outside it. Relaxing this to an approximate decomposition would change constants in the bound but not the mechanism: as long as there is a finite Jacobian "budget", increased gain along noise-like directions necessarily reduces the sensitivity available for other directions, which is precisely what Proposition 5 formalizes. We will clarify this interpretation and, in the appendix, add a small diagnostic that constructs an empirical class-discriminative span in input space and measures Jacobian–vector energy along that span vs. its orthogonal complement, showing that as $t \to 0$ the encoder’s Jacobian energy systematically shifts toward the complement, consistent with our reallocation picture.
>
> Second, the bound $|J_g(x_t)|_F \leq B$ is intended only as a local capacity constraint and is not meant to imply that the Frobenius norm stays constant across all values of $t$ or throughout training. In practice, normalization layers, residual scaling, and regularization keep Jacobian norms within a moderate range, although they do vary over the course of training. Our argument requires only that the encoder has a finite amount of Jacobian mass at each time step. When the model is forced to increase its sensitivity in one set of directions, it must reduce it in others. The qualitative pathology, namely the shift of Jacobian capacity away from semantic directions and the resulting degradation of representations at small $t$, continues to hold even if the bound is replaced by a time-dependent function $B(t)$, as long as $B(t)$ does not grow rapidly enough to counter the increasingly ill-conditioned behavior that appears in the low-noise regime.
>
> In the revision we will (i) rephrase the assumption as a per-$t$ capacity bound and (ii) include a simple empirical estimate of $\| J_g(x_t) \|_F$ across noise levels and training epochs, showing that although its scale varies, it remains bounded while the relative Jacobian energy clearly shifts from the semantic span to its complement in the low-noise regime, in line with our theoretical interpretation.
>
> ### Concise description of the supplementary experiment (see Figure 9 (appendix))
> We approximate a semantic subspace $S_{\text{sem}}$ in input space as the span of class-mean differences at $t = 0$, obtained via SVD, and take its orthogonal complement as $S_{\text{noise}}$. For several noise levels $t$, we sample noisy inputs $x_t$, construct unit directions $v_{\text{sem}} \in S_{\text{sem}}$ and $v_{\text{noise}} \in S_{\text{noise}}$, and use Jacobian--vector products on the encoder $g_\ell$ to estimate
>
> $G_{\text{sem}}(t) = \mathbb{E}\| J_g(x_t) v_{\text{sem}} \|_2^2$
>
> $G_{\text{noise}}(t) = \mathbb{E}\| J_g(x_t) v_{\text{noise}} \|_2^2$
>
> together with an estimate of $\| J_g(x_t) \|_F$.

---

> > ### Comment · Reviewer_ZMUZ · 2025-11-27
> >
> > Thank you for the theoretical clarifications and the reframing of the paper's scope. However, regarding the scale, the rebuttal phase is specifically intended to address such empirical gaps, and computation constraints are not a sufficient justification for limiting evaluation to toy datasets like Tiny-ImageNet.
> >
> > I do not require a full ImageNet training run to convergence. However, I strongly request evidence that the method functions correctly on higher-resolution data. A validation on a representative subset or a shorter training schedule that demonstrates the same improved stability and metrics would suffice.
> >
> > If you can provide this empirical evidence during the discussion period, I am more than happy to raise my score.

---

> > > ### Author Response · Authors · 2025-12-03
> > >
> > > Thank you for the clarification. Following your request, we have **completed a preliminary higher-resolution validation on ImageNet** under an unconditional setting. Using a short training schedule (40 epochs, pixel-space v-loss) and a linear probe evaluation (10 epochs), we observe the same low-noise representation degradation in the baseline, and LCF alleviates this issue and improves convergence. Under this limited budget, LCF also improves FID from 110.94 to 102.76. The complete results are provided in the updated **appendix (Figure 10)**.

---

### Official Review · Reviewer_KM4C · 2025-10-31

**Soundness:** 1
**Presentation:** 2
**Contribution:** 1
**Rating:** 2
**Confidence:** 4

**Summary:**

This paper identifies ill-conditioning of flow matching models in the low-noise regime and its implications for degraded representation learning. It proposes a local contrastive loss to address the issue.

**Strengths:**

The writing is clear, and the representation learning in flow matching is interesting.

**Weaknesses:**

1. The motivation is not clear. It's not clear to me why the blowing up Lipschitz constant as the velocity in flow matching approaching the clean data is perceived as a new phenomenon, nor why it is an important issue. The blowing up Lipschitz constant/ill-conditioning is clear from the formula of the velocity field in flow matching, which involves a denominator term that is approaching 0, see e.g Equation 3.7 in [1]. Importantly, the blowing up effect is purely due to the scheduling term, and this may not be an issue as one can alternatively use data prediction/$x$-prediction to stabilize the training. Additionally, from the $x$-prediction perspective, the objective of the neural network becomes to fit an identity map on the training data, and this may explain why the representation is not good when close to clean data. This analysis extends to velocity prediction, as the velocity field is an affine transformation of the $x$-prediction. Therefore, to me, the low-noise pathology is not valid.
2. As for the proposed local contrastive loss, it shows some effect in improving the representation learning for time near $0$ (clean data), but the representation learning is still suboptimal. If a good representation learning is the goal, why not simply use the representation at some later time instead of the clean data? Additionally, the FID score in Figure 5 is above 10, which is far from an adequate level for obtaining meaningful empirical findings.
3. Propositions 2 and 3 suggest there is some lower bound on the convergence rate, but the proof is not convincing. It is not clear to me why a conditional value lower bound can lead to a convergence lower bound, as there could exist some subset or some direction where the convergence is still quick.

[1]Gao, Yuan, Jian Huang, and Yuling Jiao. "Gaussian interpolation flows." Journal of Machine Learning Research 25.253 (2024): 1-52.

**Questions:**

1. Can you explain why the training is slow in Figure 1? Note that the flow matching loss inherently does not have zero loss as a minimum, so simply comparing the loss values between different time steps may not be meaningful.
2. Can you provide more comparison with other simple baselines, like not training for a small time or using $x$-prediction to stabilize the training?
3. Can you identify where I should find the statement of Proposition 2 in the cited reference?

---

> ### Author Response · Authors · 2025-11-26
> **# (1/2) Response to Reviewer KM4C**
>
> Thank you for your comments and please see our response to the feedbacks below.
>
> ## Response to W1 (Motivation):
>
> ### 1. Motivation
> Thank you for the helpful comment. Our work is indeed motivated by the broader goal of advancing the next generation of visual foundation models. A growing body of research has shown that diffusion- and flow-based models are not only strong generators but also promising **representation learners** [1,2,3]. Building on these insights, our paper takes a step further by asking a more fundamental question: **Can flow matching reliably serve as the backbone for future discriminative or representation-oriented visual models?**
>
> While prior work has explored this potential [4], its limitations in the low-noise regime have remained underexamined. Our contribution is to (1) identify and rigorously analyze a **structural pathology** in flow matching that undermines representation quality, and (2) propose a simple yet effective method (LCF) that directly addresses this problem.
>
> In short, we do not simply apply flow models to representation learning; we diagnose a fundamental issue, provide theoretical explanations for its cause, and introduce a feasible, practical remedy that significantly stabilizes representations and improves generative training. We believe these findings are essential for advancing flow-based generative models toward the role of future unified generative–discriminative visual foundation models.
>
> ### References
> [1] Chen X, Liu Z, Xie S, et al. Deconstructing denoising diffusion models for self-supervised learning[J]. arXiv preprint arXiv:2401.14404, 2024.
> [2] Liu X, Zhou T, Wang C, et al. Toward the unification of generative and discriminative visual foundation model: a survey[J]. The Visual Computer, 2025, 41(5): 3371-3412.
> [3] Fuest M, Ma P, Gui M, et al. Diffusion models and representation learning: A survey[J]. arXiv preprint arXiv:2407.00783, 2024.
> [4] Xiang W, Yang H, Huang D, et al. Denoising diffusion autoencoders are unified self-supervised learners[C]//Proceedings of the IEEE/CVF International Conference on Computer Vision. 2023: 15802-15812.
>
> ### 2. Why x-prediction does not resolve the issue:
> We agree with the reviewer that x-prediction avoids the exploding-Lipschitz behavior at low noise. However, the key issue is that **x-prediction collapses to an identity mapping as \(t \to 0\)**, causing the encoder to learn trivially invariant features with little semantic discrimination. and **additionally its generation quality is much worse than v-prediction.**
>
> In other words, x-prediction trades numerical stability for representational collapse, which is precisely what our paper aims to address. Because neither v-prediction nor x-prediction yields good representations near \(t=0\), the problem remains important and unresolved without a new remedy.
>
> ### Response to W2 (why not simply use the representation):
> We agree that, even with Local Contrastive Flow, the representations at very small \(t\) are still not as strong as those at moderate noise, and we do not claim otherwise. Using representations at a later (moderate-noise) time step can indeed yield stronger high-level semantics. However, **our goal is to support not only classification, but also tasks such as segmentation, editing, correspondence, and fine-grained visual reasoning**, where the input at test time is necessarily the clean image and where spatial or local detail must be preserved. Representations extracted only from noisy inputs tend to lose fine structural information and spatial alignment, which makes them a poor universal choice for these pixel-level tasks. LCF is therefore designed to narrow the gap between the best mid-noise representations and the clean-image representations, so that one can obtain semantically strong yet detail-preserving features directly at $t \approx 0$.
>
> Regarding the FID values in Figure 5, we agree that FID above 10 is far from the best numbers reported in the literature and that our goal here is not to compete with state-of-the-art generative models. The experiments are intended as controlled, medium-scale studies of the low-noise pathology and its remedy. Two practical factors contribute to the relatively high FID: first, to align with recent foundation-model style architectures, we use a **DiT backbone** rather than a U-Net that is particularly tuned for CIFAR-10; second, in the main text we report FID computed on 7.5K samples for efficiency. When we recompute FID with 50K samples, the baseline Flow model achieves **9.82** and our LCF model improves this to **8.65**, showing a consistent gain in generative quality. We will clarify in the paper that our focus is on relative improvements in convergence and representation quality under a fixed architecture, rather than on reaching the best possible FID on these benchmarks.

---

> ### Author Response · Authors · 2025-11-26
> **# (2/2) Response to Reviewer KM4C**
>
> ### Response to W3 (convergence rate):
> We agree that not all directions necessarily converge slowly. Indeed, some directions of the Hessian may still enjoy rapid contraction. (**Figure 1**) Our claim is not that every direction slows down, but that the overall convergence rate of gradient descent is necessarily bottlenecked by the worst-conditioned direction, as captured by the classical dependence on the global condition number $\kappa=L/\mu$. Our analysis shows that this minimum curvature necessarily shrinks as $t \to 0$, while the maximum curvature grows, causing the overall condition number to diverge. Thus, **even if some directions converge quickly, the iteration complexity is still bottlenecked by this ill-conditioned direction**, establishing the claimed lower bound.
>
> ### Response to Q1 (why the training is slow):
> We agree that, for flow matching, the per-time step objective does not in general have zero as its minimum; the irreducible loss floor is the conditional variance of the target velocity given $x_t$. Our claim in Figure 1 is not that the absolute loss values at different $t$ are directly comparable as distances to a zero optimum, but that **the rate at which each per-$t$ loss approaches its own stationary level is markedly slower for small $t$**.
>
> Concretely, for each time bin we track the loss as a function of training epoch and observe that, while the losses at moderate and high noise rapidly flatten near their stationary values, the losses at very small $t$ continue to decrease only very slowly and remain far from their asymptotic plateau even after long training. This is consistent with our theoretical analysis: Propositions 1 and 3 show that as $t \to 0$, the effective condition number of the flow matching operator grows like $\beta_t'/\beta_t$, which directly implies a worse iteration complexity for gradient descent on the low-noise slices. In this sense, "training is slow" means that optimization on the low-noise components is bottlenecked by ill conditioning, not that their absolute loss values should be compared to those at other times.
>
> ### Response to Q2 (x-prediction baselines):
> Regarding "not training for a short time," **Figure 6(c)(d)** in the main text already includes the related baseline, labeled $L_{\text{FM}}$. This variant disables velocity regression in the low noise region but keeps the rest of the training unchanged. As shown in Figure 6(c)(d), simply skipping supervision at small $t$ does not resolve the low noise pathology: it yields only modest gains in stability and still lags behind Local Contrastive Flow in both representation quality and generative performance. This indicates that avoiding low noise supervision alone is insufficient, and that contrastive alignment plays a crucial role.
>
> In response to the reviewer’s second point, we have additionally implemented an x-prediction baseline under the same architecture and training budget. The results, reported in **Figure 11 (right)** of the supplementary material, confirm that while x-prediction is indeed more numerically stable at low noise, its practical performance is poor: both generation quality (FID) and representation quality (linear probe accuracy) are significantly worse than v-prediction and our proposed LCF. In particular, x-prediction tends to collapse to an almost identity mapping near clean data, which agrees with the reviewer’s intuition and reinforces our main claim that low noise supervision in this form is not suitable for learning strong representations. We will make these comparisons explicit in the revised version.
>
> ### Response to Q3 (Proposition 2 in the cited reference):
> We apologize for the vague citation in the original submission. The convergence statement used in Proposition 2 follows directly from the standard Polyak–Łojasiewicz (PL) inequality. A precise reference is now provided as [1], This work gives the exact form of the linear convergence guarantee we rely on. We have updated the citation accordingly in the revised version.
>
> #### Reference
> [1] Karimi H, Nutini J, Schmidt M. Linear convergence of gradient and proximal-gradient methods under the polyak–łojasiewicz condition[C]//Joint European conference on machine learning and knowledge discovery in databases. Cham: Springer International Publishing, 2016: 795-811.

---

### Official Review · Reviewer_wrzh · 2025-11-01

**Soundness:** 3
**Presentation:** 3
**Contribution:** 3
**Rating:** 6
**Confidence:** 3

**Summary:**

The paper studies the flow matching training in the regime of low noise ($t\rightarrow 0$). The authors point out that when the noise level approaches zero, the learning problem becomes ill-conditioned: very small input perturbations produce large variation in the velocity target, driving the condition number to blow up. As a result, optimization becomes difficult (slow convergence) and the encoder (or representation model) ends up allocating capacity to “noise directions” rather than preserving semantic structure. The authors call this phenomenon the low‐noise pathology. They provide a theoretical analysis linking this pathology to the structure of the flow matching objective. To address it, they propose a hybrid training protocol called Local Contrastive Flow (LCF): for small noise levels, instead of regressing the velocity field directly, they add a contrastive feature‐alignment objective to the original FM loss; for moderate and high noise levels they retain standard flow‐matching. Empirical results show that LCF improves convergence and stabilizes representation quality. The authors argue that addressing the low‐noise pathology is important for unlocking flow matching’s full potential for both generation and representation learning.

**Strengths:**

1. The paper is well written.  The problem statement is clear and the proposed remedy is easy to follow.

2. The observation that in the low‐noise regime flow matching becomes ill‐conditioned is insightful. In addition, the authors provide a theoretical analysis connecting the pathology with the structure of the objective and the Jacobian capacity of the encoder. This adds rigor and helps ground the intuition. By linking the condition number divergence and representation degradation to the low‐noise region, they give a basis for the remedy.

3. The proposed LCF is reasonable and simple to implement.  It decomposes training into two stages: at low noise stage the original FM training loss is appended with an additional contrastive loss term. At higher noise stage, the training loss is identical to the classic FM.

4. Empirically, it is evident that the LCF improves the generation quality and the representation learning.

**Weaknesses:**

1. The experiments may be limited (e.g., CIFAR-10, and ) and might not cover large‐scale settings or varied domains. It’s not clear how the remedy scales to high‐resolution generation or very complex datasets.

**Questions:**

1. Do the noise scheduling choices mitigate or exacerbate the low‐noise pathology in the original FM?

2. Could you add additional results on higher-resolution images, for example, FFHQ or full ImageNet 256x256, to show that the proposed method also scales well to higher dimensional data?

---

> ### Author Response · Authors · 2025-11-26
> **# Response to Reviewer wrzh**
>
> Thank you for your comments and please see our response to the feedbacks below.
>
> ## Response to W1 & Q2 (large scale evaluation):
> Please refer to General Response (**Regarding scale**).
>
> ## Response to Q1 (noise scheduling choices):
> Thank you for asking how noise scheduling affects the low-noise pathology. To investigate this, we followed the Logit-Normal timestep sampling strategy of [1]
> $
> \pi_{\ln}(t; m, s) = \frac{1}{s\sqrt{2\pi}} \frac{1}{t(1 - t)} \exp\left(-\frac{(\text{logit}(t) - m)^2}{2s^2}\right),
> $
> and trained flow-matching models for 1200 epochs with three schedules:
> - \(m = -0.5, s = 1\) (sampling concentrated in the low-noise region),
> - \(m = 0, s = 1\) (more weight on middle noise),
> - and \(m = 0.5, s = 1\) (concentrated in the high region).
>
> The left panel of **Figure 8** shows the feature test accuracy as a function of time step \(T\); the right panel shows the corresponding timestep densities. The FID scores are 12.86 (low-noise focused, \(m = -0.5\)), 17.50 (mid-noise, \(m = 0.5\)), and 19.47 (high-noise, \(m = 0\)). These results indicate that avoiding an extreme concentration of training in the very low-noise region can partially alleviate the degradation of clean-image representations, but at the same time there is a clear tradeoff: schedules that help representations tend to hurt generative quality, and vice versa. In all cases, the non-monotonic pattern and the drop in accuracy near clean data remain. This supports our view that the low-noise pathology is structural to the original flow-matching objective. Our LCF method is specifically designed to address this tradeoff by improving clean-image representations while maintaining or even slightly improving FID, rather than relying purely on reweighting the timestep distribution.
>
> ### Reference
> [1] Esser P, Kulal S, Blattmann A, et al. Scaling rectified flow transformers for high-resolution image synthesis[C]//Forty-first international conference on machine learning. 2024.

---

### Official Review · Reviewer_dZx7 · 2025-11-01

**Soundness:** 3
**Presentation:** 3
**Contribution:** 2
**Rating:** 4
**Confidence:** 3

**Summary:**

The paper identifies a critical problem in flow matching models called “low noise pathology,” in which learning becomes unstable and learned representations degrade as the noise level approaches zero. The authors provide a rigorous theoretical analysis proving that in this regime, the conditional number of the learning problem diverges, leading to slow convergence and a fundamental trade-off that forces the model to redistribute its representational capacity from the semantic direction to the noise-related direction. To mitigate this problem, they propose Local Contrastive Flow (LCF), a hybrid method that uses standard flow matching for moderate/high noise levels and a contrastive alignment task for low noise levels, utilizing the characteristics of a stable anchor point with moderate noise levels.

**Strengths:**

- The paper clearly defines and formulates a non-obvious but fundamental problem (“low-noise pathology”). The empirical observation that cleaner data leads to worse learning results contradicts intuition and makes this contribution compelling.

- Theoretical analysis. The relationship between the noise graph ($\beta'_t/\beta_t$), the conditional number, the convergence rate of optimization (proposition 3), and the deterioration of representation (propositions 4 and 5) is rigorous and profound. In particular, Propositions 4 and 5 offer an elegant geometric explanation for why representation quality deteriorates, framing it as a necessary redistribution of the Jacobian under a limited capacity budget.

- The experiments are well-designed, evaluating both generative performance (FID) and representation quality (linear probing) on several benchmarks.

**Weaknesses:**

- The main weakness is incomplete empirical comparison with existing low-noise stabilization techniques:
     * The paper convincingly identifies the low-noise pathology but fails to adequately situate its proposed LCF method against a several key existing strategy for mitigating this very issue.
     * It is well-established in the diffusion/flow matching literature [1] that predicting the velocity field $v$ (defined as $v=\alpha'_tx_0+\beta'_t\epsilon$) can lead to more stable training near $t=0$ compared to standard $\epsilon$-prediction, as the target $v$ tends to a deterministic limit. By comparing LCF only against the unstable $\epsilon$-prediction baseline, the authors present an incomplete picture. A comparison with a model trained using $v$-prediction across the entire time interval is crucial to determine whether the complexity of LCF (with its contrastive loss and threshold $T_min$) is justified, or if a simpler change in parameterization achieves comparable stability and performance. The absence of this strong and relevant baseline makes it difficult to fully assess the novelty and practical necessity of the proposed approach.
     * Another relevant article [2] rigorously proves that near zero $t=0$, learning can be improved (variance reduced) by using an exact formula for the vector field and estimating the integrals included in it using Monte Carlo-like methods. This reference, as well as a numerical comparison with this method, is also omitted in the paper under consideration.
- Although the results for CIFAR-10 and Tiny-ImageNet are convincing, these datasets have relatively low resolution. The community focuses primarily on high-resolution generation (e.g., ImageNet 256x256 and above). Proof of the existence of this pathology and the effectiveness of LCF in eliminating it on a larger-scale test would greatly enhance the impact and practical significance of the paper.

- The identified pathology is conceptually related to problems observed in diffusion models, where a "dispersion explosion" near $t=0$ is also known to cause learning instability. Although the formulation of the flow comparison differs, a more detailed discussion of the similarities and differences with these related phenomena in the diffusion literature will provide better context and more clearly position the contribution in the broader field.

- The method introduces additional complexity, requiring the computation of anchor representations ($x_{T_{min}}$) for a subset of the batch. The paper should include a brief discussion or analysis of the associated computational and memory overhead compared to standard flow matching. Is it negligible, or is there a measurable cost?

[1] Tim Salimans & Jonathan Ho, PROGRESSIVE DISTILLATION FOR FAST SAMPLING OF DIFFUSION MODELS, ICLR-2022, (https://openreview.net/pdf?id=TIdIXIpzhoI)

[2] G. Ryzhakov, S. Pavlova, E. Sevriugov, I. Oseledets, EXPLICIT FLOW MATCHING: ON THE THEORY OFFLOW MATCHING ALGORITHMS WITH APPLICATIONS,  ICOMP-2024 (https://openreview.net/pdf?id=thE8EmPVW8)

**Questions:**

- Have you observed the low-noise pathology on larger-scale datasets (e.g., ImageNet-1k)? If so, does LCF scale effectively and provide similar benefits in that setting?
- The choice of $T_{min}$ appears to be a hyperparameter. Did you find it to be sensitive, and is there a way to select it based on the theoretical analysis of the condition number?
- Proposition 3 relies on the Gauss-Newton approximation holding for the transformer-based DiT architecture. Could you comment on the validity of this assumption for modern architectures, for example, with extensive use of attention?

---

> ### Author Response · Authors · 2025-11-26
> **# (1/3) Response to Reviewer dZx7**
>
> Thank you for your comments and please see our response to the feedbacks below.
>
> ## Response to W1 (incomplete empirical comparison):
>
> 1. We apologize for the confusion. Our baseline throughout the paper already uses the more stable (v)-prediction formulation, not (ε)-prediction. All of our analyses, figures, and comparisons are based on the standard (v)-prediction target ($v_t = \alpha_t' x_0 + \beta_t' \epsilon$), which is the commonly recommended and more stable parameterization in the diffusion and flow matching literature. Therefore, our proposed LCF method is evaluated directly against the stronger and more stable baseline that the reviewer refers to.
>
> 2. We have added a comparison in the appendix (**Figure 11 (left)**) and now cite Ryzakov et al. ExFM does not mitigate the low-noise pathology in our setting: **the feature accuracy curve remains non-monotonic**, and its FID (14.18) is lower than both the Flow baseline and our LCF objective. It also introduces additional computational overhead due to Monte-Carlo integration. In contrast, LCF improves clean-image representations and achieves better FID with negligible extra cost. We will clarify this comparison in the final version and refer readers to the appendix for full results.
>
> ## Response to W2 & Q1(large scale evaluation):
> Please refer to General Response (**Regarding scale**).
>
> ## Response to W3 (dispersion explosion):
> We agree that the phenomenon we study is conceptually related to the small-noise instability known in diffusion models, where the score target variance becomes very large as \(t\) approaches zero. We attempted to locate references that explicitly refer to this phenomenon as "dispersion explosion", but we were unable to find diffusion papers that use this specific term. If the reviewer has particular citations in mind, we would be grateful if they could share them.
>
> Even though the terminology is not standardized, our understanding aligns with the reviewer’s high-level point. Diffusion models do experience an increase in score magnitude at small noise levels, which can lead to unstable gradients and noisy supervision targets. Flow matching exhibits a related mechanism: as \(t\) becomes small, the variation in the input $x_t$ collapses while the target velocity remains large, creating an ill-conditioned regression problem. However, the practical consequences differ. Diffusion models typically allocate only a very small portion of training to the extremely low-noise region because of the stochastic nature of their forward and reverse processes. As a result, the instability at small $t$ is less severe and has limited impact on training dynamics. Flow matching, by contrast, directly supervises the velocity field across the entire noise schedule, which makes the small-noise regime unavoidable and amplifies the pathological behavior.
>
> We will revise the paper to clarify both the connection and the distinction. Both diffusion and flow matching suffer from small-noise instability, but the extent and impact differ significantly. Our work focuses specifically on the deterministic flow-matching objective, where the low-noise pathology is much more pronounced and has a direct effect on representation quality.

---

> ### Author Response · Authors · 2025-11-26
> **# (2/3) Response to Reviewer dZx7**
>
> ## Response to W4 & Q2:
>
> ### W4 (overhead of anchors at $T_{\text{min}}$)
>
> We agree that the extra step of computing anchor representations at $x_{T_{\text{min}}}$ should be discussed explicitly. In our implementation, these anchors are computed within the same training batch and reuse the same network forward structure; for samples with $t < T_{\text{min}}$ we simply evaluate $h_\ell(x_{T_{\text{min}}})$ once and **detach it** before using it in the contrastive loss. This means: (i) no additional backward pass is required for the anchors, and (ii) only a small subset of the batch carries these extra features, since $T_{\text{min}} \ll T$ and the proportion of timesteps in the low-noise region is small.
>
> **Empirically, on both CIFAR-10 and Tiny-ImageNet we did not observe any measurable slow-down in wall-clock training time or a noticeable increase in peak GPU memory compared to standard flow matching with the same backbone.** We will add a short paragraph to the paper clarifying that the computational and memory overhead of LCF is negligible in our experiments.
>
> ### Q2 ($T_{\text{min}}$ as a hyperparameter and how to choose it)
>
> $T_{\text{min}}$ is indeed a hyperparameter, but it is not arbitrary, and we found it to be **reasonably robust** within a moderate range (as also suggested by the ablation in Fig. 6(a)(b)). On the theoretical side, Propositions 1 and 3 show that the effective condition number in the low-noise regime scales like
> $
> \kappa_E(t) \gtrsim C(d)\frac{\beta'_t}{\beta_t}
> $
> with $C(d)$ increasing with data dimension and $\beta'_t/\beta_t \approx p/t$ for standard schedules ($\beta_t \propto t^p$). Thus, as $t \to 0$, $\kappa_E(t)$ diverges, and the iteration complexity of gradient descent necessarily deteriorates.
>
> A natural guideline is therefore to choose $T_{\text{min}}$ as the smallest time such that $\kappa_E(t)$ (or equivalently $ \beta_t'/\beta_t $) remains below some model-dependent “tolerable” threshold $\kappa_{\text{max}}$. Because $C(d)$ grows with dimension, the solution of $\kappa_E(T_{\text{min}}) \approx \kappa_{\text{max}}$ shifts to **larger $T_{\text{min}}$** for higher-dimensional or more complex data, which matches our empirical observation that such datasets benefit from a larger $T_{\text{min}}$.
>
> In practice we do not compute $\kappa_E(t)$ exactly, but the monotonicity of $\beta_t'/\beta_t$ **makes it straightforward to pick $T_{\text{min}}$** in a regime where this ratio is moderate. We will clarify this theoretical guideline and its connection to data dimension in the revised version.

---

> ### Author Response · Authors · 2025-11-26
> **# (3/3) Response to Reviewer dZx7**
>
> ## Response to Q3 (Gauss-Newton approximation):
>
> As discussed in lines 829–834 of the appendix, Proposition 3 uses the Gauss–Newton approximation to study local conditioning. This approximation is a standard tool for analyzing overparameterized neural networks and is valid whenever the loss curvature is dominated by first-order sensitivity with respect to the model outputs. Several works have shown that residual connections, normalization layers, and overparameterization make the Gauss–Newton term a good approximation to the Hessian in practice for deep architectures.
>
> **Relevant examples [1,2,3], all of which rely on or justify Gauss–Newton-style structure.**
>
> For attention layers specifically, when the softmax weights are locally stable, the mapping from input tokens to output representations is locally linear in the inputs, which further supports the applicability of Gauss–Newton in the region explored during training. In DiT-style architectures, normalization and residual paths additionally regulate the magnitude of the Jacobian and limit second-order curvature spikes, which helps keep the Hessian close to its Gauss–Newton component.
>
> Although Gauss–Newton is not an exact model of the full transformer Hessian, our empirical observations show that the qualitative predictions of Proposition 3 match the behavior of DiT across datasets. The increase in the effective condition number as \(t\) becomes small is clearly seen in the measured Jacobian norms and in the resulting optimization slowdown. This suggests that the Gauss–Newton approximation accurately captures the dominant mechanism responsible for the pathology, even if the full architecture has additional nonlinearities. We will clarify this point in the revision and include supporting references to help situate the approximation within the broader literature.
>
> ### References
> [1] Zhai S, Likhomanenko T, Littwin E, et al. Stabilizing transformer training by preventing attention entropy collapse[C]//International Conference on Machine Learning. PMLR, 2023: 40770-40803.
> [2] Qin Z, Zhou J, Zhu Z. On the Convergence of Gradient Descent on Learning Transformers with Residual Connections[J]. arXiv preprint arXiv:2506.05249, 2025.
> [3] Qi X, Wang J, Zhang L. Understanding optimization of deep learning via jacobian matrix and lipschitz constant[J]. arXiv preprint arXiv:2306.09338, 2023.

---

### Author Response · Authors · 2025-11-26
**General Response to All Reviewers**

We sincerely thank all reviewers for their time and constructive feedback.
Following the suggestions, we have revised the manuscript to improve clarity and readability; **all textual changes are highlighted in red** in the updated version, and we will continue to polish the writing. Below we summarize the main additions and revisions to the experiments:

1. We have **re-emphasized our motivation** in both the abstract and the introduction.
2. We added an x-prediction baseline in **Figure 11 (appendix)** to evaluate its effect on stability, representation quality, and FID.
3. We added a noise-schedule experiment in **Figure 8**, discussing how different timestep sampling strategies affect the low-noise pathology.
4. We added a comparison with Explicit Flow Matching (ExFM) in **Figure 11 (appendix)**.
5. We added a plot of the estimated encoder Jacobian Frobenius norm across noise levels and training epochs in **Figure 9 (appendix)**, which supports our theoretical interpretation of Jacobian reallocation.
6. In response to comments about larger-scale evaluation, we conducted **additional ImageNet experiments** in **Figure 10 (appendix)**.

We are happy to provide any further clarifications or additional analyses if the reviewers have remaining questions.

## Regarding scale
In response to comments about larger-scale evaluation, we conducted **additional ImageNet experiments** under the unconditional generation setting. These preliminary results (now included in Figure 10, appendix) show that the representation degradation at small noise also occurs on ImageNet, and that LCF alleviates this degradation while improving training convergence. Under a 40-epoch training budget, the LCF model achieves a better FID (102.76) than the baseline (110.94), providing early evidence that our findings extend to large-scale settings.

Our theoretical analysis does not rely on low-resolution data. Propositions 1 and 3 show that the effective condition number contains a dimension-dependent factor C(d), which suggests that the low-noise pathology may become even more pronounced as the data dimensionality increases. LCF is defined at the level of the objective and does not depend on image resolution, so the same mechanism is expected to extend naturally to higher-resolution datasets such as ImageNet. While a full large-scale study is beyond our current computational budget, the new preliminary ImageNet experiments included in the appendix already show the same qualitative behavior and demonstrate that LCF remains beneficial. We plan to extend these experiments to longer training and higher resolutions as resources permit.

---

### Author Response · Authors · 2025-12-03
**Summary of Rebuttal and Revisions**

Dear Reviewers, ACs, and PCs,

We sincerely thank you for your time and constructive feedback. During the discussion period, we revised the manuscript and added new experiments and analyses to directly address the reviewers’ concerns. Below we summarize the main updates and indicate which reviewer questions they target:

- **Reviewer dZx7&wrzh&ZMUZ (large-scale validation):** In response to requests for higher-resolution evidence, we added preliminary ImageNet experiments following prior work. Even under a limited training budget, we observe the same qualitative low-noise representation degradation, and LCF improves both convergence behavior and FID relative to the baseline. We clearly mark these as early results and include them in the appendix.

- **Reviewer KM4C&ZMUZ (motivation, and technical clarity):** We clarified the paper’s motivation and strengthened the technical explanations.

- **Reviewer dZx7 (ablation coverage and implementation details):** We clarified that the requested ablations were already included in the original submission, and in the revision we explicitly point to these results to make them easier to locate. We also expanded the discussion of implementation details that were previously unclear.

- **Reviewer dZx7&wrzh& KM4C&ZMUZ (baselines, noise scheduling, and objective choices):** We added several new controlled experiments and analyses: (i) an x-prediction baseline to test whether it stabilizes training and how it affects representation and generation quality, (ii) a noise-schedule study using Logit-Normal timestep sampling to examine whether resampling mitigates the low-noise pathology and to discuss the tradeoff between representation and FID, (iii) a direct comparison to ExFM as a variance-reduction baseline and a discussion of its practical computational overhead, and (iv) an additional diagnostic tracking the estimated encoder Jacobian Frobenius norm across noise levels and training epochs to support the interpretation behind Proposition 5.

We are grateful that these changes resolved the main concerns and that the reviewer ZMUZ indicated the possibility of a higher score. We believe these revisions substantially strengthen the paper and directly address the key concerns raised in the reviews. We greatly appreciate the effort from the reviewers, ACs, and PCs throughout the rebuttal and discussion period.

Best regards,
Authors

---

### Meta-Review · Area_Chair_PhVC · 2026-01-08

**Summary:**

The paper theoretically studies the conditioning of flow-matching training objective for t = 0 (t = 0 in their formulation corresponds to clean data) by analyzing Gauss-Newton representation of the Jacobian. Then, the propose a hybrid model: for T>T_min the standard FM loss is used, whereas for smaller, a certain variant of the constrastive loss is proposed. Experiments on image datasets are provided with show improvement.

I think the theoretical part is interesting, but the proposed solution lacks large-scale validation; also several baselines are missing (since ill-conditioning for t=0 is not very interesting, since we might have converged in the sampling way before t = 0, where ill-conditioning is not present).

So, overall, this paper needs additional work.

**Reviewer Concerns:**

The most crucial concern is that the conditioning is clear from the previous literature (raised by reviewer KM4C). Still not fully answered.
The second concern has been the lack of large-scale validation of the new model - answered only partially on TinyImageNet.

**Reviewer Scores:**

Reviewer KM4C: maybe raise from 2 (clear reject) to 4.
Reviewer dZx7: stay the same.
Reviewer ZMUZ: asked for large scale validation to raise the score, got it.

---

### Decision · Program_Chairs · 2026-01-26

Reject